# Unraveling enteroendocrine cell lineage dynamics and associated gene regulatory networks during intestinal development

Sara Jiménez[1,2,3,4], Florence Blot[1,2,3,4], Aline Meunier[1,2,3,4], Rishabh Kapoor[1,2,3,4], Valérie Schreiber[1,2,3,4], Colette Giethlen[1,2,3,4], Sabitri Ghimire[1,2,3,4], Maxime M. Mahe[5,6], Nacho Molina[1,2,3,4,*], Adèle De Arcangelis[1,2,3,4,*] and Gérard Gradwohl[1,2,3,4,*]

**ABSTRACT**

Enteroendocrine cells (EECs) are rare intestinal epithelial cells producing multiple hormones that regulate essential aspects of digestion and energy. EEC subtypes, their hormone repertoire and differentiation mechanisms from intestinal stem cells have been characterized in the adult intestine. Although EECs must be functional from birth because their absence leads to severe intestinal malabsorption in newborns, the processes that determine their subtype specification during development remain largely unknown. We used mouse embryos, human pluripotent stem cell-derived intestinal organoid models and single-cell transcriptomics to characterize EEC lineages and dynamics during development. Our findings demonstrate that in both mice and humans, the majority of EECs are specified during development through similar differentiation trajectories to those observed in the adult intestine. This suggests that EEC subtype specification occurs independently of fully organized crypt-villus structures and stimulation by diet or microbiota. However, the emergence of certain EEC subtypes depends on tissue maturation. Finally, our integrative approach infers lineage-specific regulators dynamically, identifying new candidates controlling EEC differentiation in the developing human gut.

KEY WORDS: Stem cell, Cell fate, Organoid, Enteroendocrine, Gene regulatory networks, Hormone

## INTRODUCTION

Enteroendocrine cells (EECs) play a crucial role in maintaining intestinal homeostasis and energy metabolism, by regulating various aspects of digestion, whole-body metabolism, glycemia, food intake, appetite and weight gain. They constitute a population of rare heterogeneous cells, singly scattered and unevenly distributed over the entire intestinal mucosa. EECs sense and respond to stimuli either from the gut lumen – dietary nutrients, microbial metabolites – or from the submucosa, where they receive nerve, hormonal or immune signals. Subsequently, they secrete various hormones that either act locally or remotely through the bloodstream targeting organs like the pancreas, the stomach or the brain (reviewed in Gribble and Reimann, 2019).

Originally, EECs were categorized based on their primary hormone secretion, encompassing: enterochromaffin (EC) cells (Serotonin/5-HT), D cells (Somatostatin, SST), I cells (Cholecystokinin, CCK), K cells (Gastric inhibitory peptide, GIP), L cells (Glucagon like peptide 1, GLP-1), M cells (Motilin, MLN), N cells (Neurotensin, NTS), S cells (Secretin, SCT), and X/A cells (Ghrelin, GHRL). Studies in mice have provided evidence that EECs express a spectrum of different hormones (Egerod et al., 2012; Habib et al., 2012) such as Tachykinin (Substance P, Tac1) and Secretin in EC cells; and GLP-2 and PYY (peptide YY) in L cells. Hormone co-expression has been confirmed and further characterized by single-cell transcriptomics (Gehart et al., 2019; Grün et al., 2015; Haber et al., 2017; Piccand et al., 2019). It is believed that cellular hormone diversity reflects hormonal plasticity in response to extrinsic stimuli rather than distinct lineages (Beumer et al., 2018; Gehart et al., 2019). Single-cell transcriptomic analyses have revealed that, in the adult intestine, EECs differentiate from crypt-based Neurog3 endocrine progenitors along two primary trajectories: one leading to EC cells expressing serotonin (5-HT), the other to peptide (PE) hormone-expressing EEC subtypes (including GHRL, GIP, GLP-1, SST) (Gehart et al., 2019; Piccand et al., 2019). Despite a few loss-of-function studies in mice uncovering the roles of Neurog3 downstream transcription factors (e.g. Arx, Pax4, Foxa1/2, Insm1, NeuroD, Rfx6, Isl1, Lmx1a, Nkx2.2) in specifying different EEC subtypes (Beucher et al., 2012; Du et al., 2012; Gierl et al., 2006; Gross et al., 2016; Naya et al., 1997; Piccand et al., 2019; Terry et al., 2014; Ye and Kaestner, 2009), the gene regulatory networks governing EEC diversity remain largely unknown. Thus, essential aspects of EEC differentiation have been deciphered mainly using mouse models. Human studies have long been hampered by the scarcity of EECs and the difficulty of obtaining human tissue. Major advances have been achieved through to the development of relevant *in vitro* models such as intestinal 3D enteroids or 2D cultures derived from human adult intestinal stem cells (Fujii et al., 2018; Singh et al., 2024; Zeve et al., 2022). Notably, these models have allowed the generation of single cell mRNA atlas and establish the secretome of purified human mature EEC subtypes but mostly relied on the overexpression of NEUROG3 to induce and boost EEC differentiation (Beumer et al., 2020).

In contrast to the adult intestine, the diversity, differentiation dynamics and lineage trajectories of EECs during intestinal morphogenesis remain poorly understood. This knowledge is particularly important given that EECs must be functional at birth.

[1]Université de Strasbourg, IGBMC UMR 7104- UMR-S 1258, Developmental Biology and Stem Cells Department, F-67400 Illkirch, France. [2]CNRS, UMR 7104, F-67400 Illkirch, France. [3]INSERM, UMR-S 1258, F-67400 Illkirch, France. [4]IGBMC, Institut de Génétique et de Biologie Moléculaire et Cellulaire, F-67400 Illkirch, France. [5]Nantes Université, CHU Nantes, INSERM, TENS, The Enteric Nervous System in Gut and Brain Diseases, IMAD, F-44000 Nantes, France. [6]Center for Stem Cell and Organoid Medicine, Cincinnati Children's Hospital Medical Center, Cincinnati, OH.

*Authors for correspondence (gradwohl@igbmc.fr; adele@igbmc.fr; nacho.molina@igbmc.fr)

S.J., 0000-0001-7103-7474; A.M., 0000-0003-3594-168X; V.S., 0000-0003-0507-639X; S.G., 0000-0001-6992-1148; M.M.M., 0000-0003-2560-193X; N.M., 0000-0003-0233-3055; A.D., 0000-0003-1114-8441; G.G., 0000-0002-6730-2615

Studies on both mice and humans have shown that failures in EEC development can lead to chronic and potentially fatal malabsorptive diarrhea in neonates (Blot et al., 2023; Mellitzer et al., 2010; Wang et al., 2006). Hence, the onset of EEC specification occurs during fetal development, coinciding with intestinal morphogenesis, which takes place throughout the first and second trimesters in humans, unlike in mice, where it is completed 2 weeks after birth. In mice, EECs begin to emerge around E14.5, preceding the establishment of bona fide crypt-villi structures (Sprangers et al., 2021). Human intestinal development can be investigated through pluripotent stem cell derived human intestinal organoids (HIOs), which resemble the fetal intestinal epithelium and, unlike crypt derived enteroids, contain also mesenchyme, that can serve as an epithelial stem cell niche (Poling et al., 2024; Spence et al., 2011). When transplanted into mice (referred to as tHIOs), HIOs undergo maturation (e.g. formation of crypt–villi axis), rendering them an excellent model for studying human intestinal development (Finkbeiner et al., 2015; Singh et al., 2023; Watson et al., 2014). HIOs contain EECs, which depend on the proendocrine transcription factor NEUROG3, thus capturing the very early stages of EEC specification observed during fetal development (Spence et al., 2011). Notably, Sinagoga and colleagues showed that HIO transplantation allowed the development of all known EEC subtypes (Sinagoga et al., 2018). However, EEC differentiation was enhanced by NEUROG3 overexpression, which may alter the transcriptional interactions underlying cell fate decisions during EEC subtype specification. Moreover, the gene regulatory network governing human EEC differentiation remains poorly studied.

This study is centered on characterizing EEC differentiation trajectories and the gene regulatory networks (GRNs) that regulate their diversity during development in both mice and humans. We used genetically modified mouse embryonic intestines and HIOs/ tHIOs models to isolate and study EEC lineages. We combined advanced scRNA-seq analyses with FateCompass, a computational pipeline that infers lineage-specific regulators dynamically (Jiménez et al., 2023), to reveal the cell trajectories and GRNs driving EEC diversification in the developing human intestine. This investigation offers novel insights into the differentiation trajectories of EECs and the molecular mechanisms underlying human EEC heterogeneity in the developing intestine.

## RESULTS
### Enteroendocrine cell diversification occurs prior to birth in the immature mouse small intestine
To dissect the mechanisms governing EEC differentiation during development, we used the Neurog3[eYFP] mouse model, which allows the tracing and purification of all eYFP+ EECs from Neurog3-expressing enteroendocrine progenitors to various differentiated EEC subtypes (Blot et al., 2023; Mellitzer et al., 2004; Piccand et al., 2019). We used a plate-based scRNA-seq technology (Sorting and Robot assisted Transcriptome sequencing, SORT-seq, Muraro et al., 2016) to profile eYFP+ EECs isolated at embryonic day 15.5 (E15.5), approximately 1 day after the emergence of the first EECs and the onset of villus formation; at E18.5, nearing the completion of villi elongation, and prior to birth and exposure to food and microbiota; and at postnatal day 12 (P12), following crypt formation and preceding weaning (Fig. 1A). We analyzed the transcriptional profile of a total of 1138 cells, comprising 384 from E15.5 small intestine, 514 from E18.5, and 240 from P12 retaining 14,495 genes. Uniform manifold approximation and projection (UMAP) embedding revealed the interweaving of cells from the different stages reflecting analogous transcriptional profiles (Fig. 1B). Actively cycling cells were

observed from the P12 stage (Fig. 1C), corresponding primarily to undifferentiated progenitor cells (see below), which may be associated with crypt development. Unsupervised graph-based clustering of the 1138 eYFP+ cells revealed 11 major cell clusters named based on the expression of their main hormonal products as well as known markers (Fig. 1D,E). These clusters include EEC progenitors with strong *Neurog3* expression (Neurog3 Progenitors), peptide hormone progenitors (PE progenitors) and five clusters of peptide hormone expressing cells, KIS (*Gip+, Cck+, Sct+*), L (*Gcg+, Pyy+*), LINS (*Gcg+, Pyy+, Cck+, Nts+, Sct+*), X (*Ghrl+*), DX (*Sst, Ghrl+*), and two clusters of EC cells, EC early (low *Tac1*) and EC (*Tph1+, Chga+*). An additional cluster, named PE[low], was identified with very few differentially expressed genes and low expression of several PE hormone genes (including *Sct, Gcg, Ghrl, Sst, Nts,* and *Cck*). Pseudotemporal ordering of the cells (Fig. 1F; Fig. S1) suggests that EECs differentiate from progenitor populations along two main branches, the EC and the PE branch. The PE branch is composed of five clusters: X cells, DX cells, L cells, LINS cells and KIS cells. The EC branch is composed of two clusters, EC early (low *Tac1*) and EC (high *Tph1*). Of note, PE and EC clusters are characterized respectively by the expression of the transcription factors *Isl1* and *Lmx1a* (Fig. 1E), as shown in adults (Gehart et al., 2019). Interestingly, the PE lineage further divides into two sub-branches: one including L/LINS cells (*Gcg, Cck, Nts, Pyy, Sct*) and the other KIS cells (*Gip, Cck, Sct*) (Fig. 1F). Importantly, similar EECs clusters and differentiation trajectories were observed in the adult mouse intestine, suggesting that EECs might differentiate following similar paths in both the developing and adult mouse intestine. Most peptide hormone expressing cells were present as early as E15.5 (Fig. 1G; Fig. S2). Intestinal development from E15.5 to E18.5 was accompanied by significantly higher expression levels of peptide hormone genes (*Ghrl, Gcg, Pyy, Cck, Sst, Sct*) and the emergence of cells expressing *Nts* and *Gip* (Fig. 1G). Immature (*Tac1+*) EC cells were found at E15.5 and later expressed *Tph1* encoding Tryptophan Hydroxylase required for serotonin (5-HT) synthesis at E18.5 (Fig. 1G).

In summary, these findings suggest that the diversity of enteroendocrine cells (EECs) in the embryo follows analogous differentiation trajectories from those observed in the adult intestine. Notably, this process does not necessitate crypt-villus organization or stimulation from luminal contents such as nutrients or microbiota.

### Human intestinal organoids mimic small intestine enteroendocrine cell differentiation
We next investigated the diversity of developing human EECs. We opted not to employ any methods to enhance EEC formation (such as inhibiting the NOTCH pathway or inducing NEUROG3 expression), in order to avoid interfering with the GRNs that control the natural processes. Our strategy was to recapitulate intestinal development, *in vitro*, by generating HIOs from pluripotent stem cells in which EEC lineages can be traced and purified (Fig. 2) (Schreiber et al., 2021). Analysis of HIO, generated with the NEUROG3-HA-P2A-Venus iPSC line showed that Venus marks enteroendocrine progenitors and hormone-expressing EECs as expected (Fig. S3A-D; Movie 1). Venus-positive EECs were sorted from HIOs or transplanted HIOs (tHIOs) under the kidney capsule of recipient mice, to promote tissue maturation, and sequenced using plate-based scRNA-seq SORT-seq (Fig. 2) (Muraro et al., 2016). For HIOs, we profiled 633 cells (Fig. 3), from which 156 cells (∼24%) expressed *NEUROG3*. Unsupervised graph-based clustering, followed by the study of marker gene expression and proliferation status, revealed six major cell clusters, including Proliferative progenitors, Progenitors, NEUROG3 progenitors, Peptidergic EEC (XMD-LK), EC early,

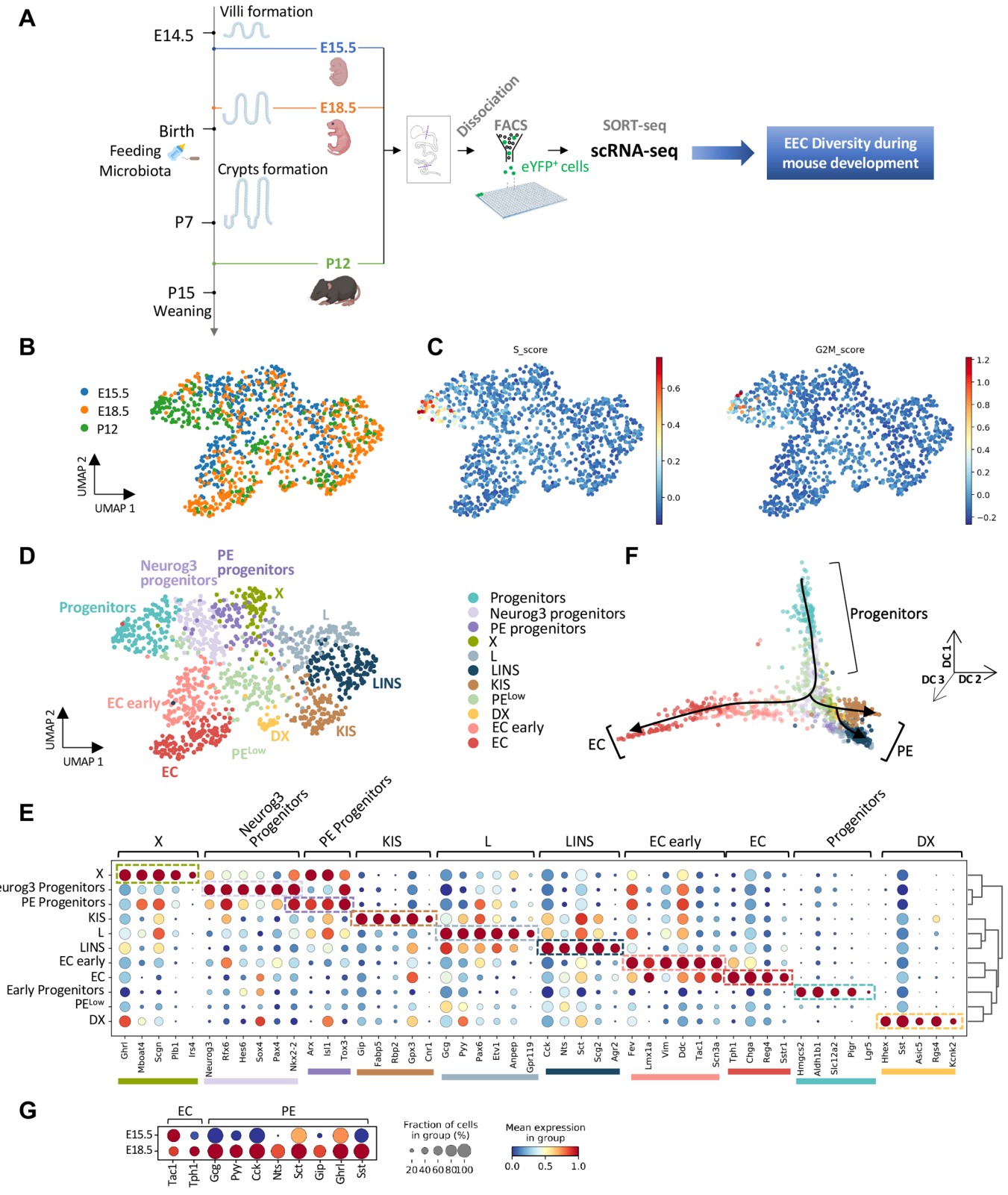

**Fig. 1.** See next page for legend.

and EC late (Fig. 3A). Approximately 65% of the population consisted of hormone-producing cells (414 cells) (Fig. 3B,E), representing distinct EECs at various differentiation stages.

The smallest cluster was Proliferative progenitors (Fig. 3B,C) expressing *NEUROG3* at low levels (Fig. 3D) and proliferative

markers such as *TOP2A* and *PCNA* (Fig. 3F), in agreement with previous studies in the mouse adult small intestinal epithelium reporting that one-third of NEUROG3+ cells are slowly cycling progenitors (Bjerknes and Cheng, 2006). NEUROG3 progenitors cluster had the highest expression of *NEUROG3* (Fig. 3D) and the

**Fig. 1. Characterization of EEC diversity in the mouse embryonic and post-natal intestine by single cell transcriptomics.** (A) Experimental design strategy used to study EEC diversity during mouse development. Small intestines from Neurog3eYFP/+ mice were collected at E15.5 (n=18), E18.5 (n=29) and P12 (n=7). After dissociation, the DAPI−/eYFP+ cells were sorted by FACS and collected in a 384-well plate (1 cell/well), and processed using SORT-Seq technology (Single Cell Discoveries) to determine transcriptomic signatures. Bioinformatics analyzes enable to identify EEC subtypes and to establish cell lineage trajectories. (B) UMAP plot showing the embedding of 1138 profiled EECs from E15.5 (384 cells), E18.5 (514 cells) and P12 (240 cells). (C) UMAP plot indicating cell cycles scores, namely the S and G2/M scores. (D) UMAP embedding of EECs throughout mouse development (E15.5, E18.5 and P12). Colors highlight clustering into 11 clusters, including progenitors and hormone-producing cells. Color code defined in D was used in panels D, and F. (E) Dot-plot showing expression of known and uncharacterized cell type-specific gene sets of EECs per cluster. The hierarchical clustering is represented on the right. Color intensity indicates mean expression (normalized) in a cluster; dot size indicates the proportion of cells in a cluster expressing the gene. (F) Diffusion map plots depicting the clusters in 3D. (G) Dot-plot representing the expression of hormone coding genes per stage. Color intensity indicates mean expression (normalized) in a cluster; dot size indicates the proportion of cells in a cluster expressing the gene.

Venus reporter; it also expressed *RFX6* (Fig. 3F), which, as we previously observed in the mouse, is expressed in endocrine progenitors and hormone expressing EECs (Piccand et al., 2019). The other three clusters represent cells expressing hormone-encoding genes. Two subsets of Enterochromaffin cells that expressed markers of EC cells (*FEV, DDC, CRYBA2, SLC18A1* including *GCH1* encoding GTP cyclohydroxylase essential for serotonin synthesis (Fig. 3F). The third subset of cells, called XMD-LK, expressed different peptide hormone encoding genes (*GHRL+, MLN+, SST+, GCG+, GIP+*). *GHRL* was highly expressed in this cluster (Fig. 3E,F). Notably, *MLN* is expressed in EECs only in the human proximal small intestine, further validating the *in vitro* differentiation of hiPSCs into human intestinal epithelial cells. Some other peptide-like hormones were detected at lower levels in the XMD-LK cluster (*GIP, GCG, PYY*) (Fig. 3E,F). Figs S4 and S5 summarize the top 50 differentially expressed (DE) genes in each cluster. EC clusters expressed *CHGA*, a prototypical EC marker in mice and humans (Goldspink et al., 2018) and *SLC18A1*, involved in serotonin transport (Lohoff et al., 2006).

EC early cells were characterized by the expression of the transcription factor *PAX4*, which is essential for serotonin cell differentiation in the mouse (Beucher et al., 2012) and human (Lin et al., 2023). In contrast, EC late cells predominantly expressed *LMX1A* (Fig. 3D,F), which is known to regulate *TPH1*, the limiting enzyme involved in serotonin synthesis, in the mouse (Gross et al., 2016). Finally, XMD-LK cells expressed both *RFX6*, *ISL1* and *ARX* (Fig. 3D,F), which are known to promote the differentiation of peptide-secreting EECs in the mouse (Beucher et al., 2012; Piccand et al., 2019). In addition to known markers, many novel and functionally uncharacterized genes were identified at different stages of EEC fate specification. For example, we found that *AMOTL2* (*Angiomotin like 2*) (Fig. S4) was enriched in the NEUROG3 progenitor cells. Members of the angiomotin family are important for tight junction formation and cell polarity and AMOTL2 has been shown to interact with and inhibit YAP (Hong et al., 2016). Interestingly, during pancreas development, Scavuzzo et al. demonstrate that *AMOTL2* loss of function decreased the number of insulin (*INS+*) cells while increasing glucagon (*GCG+*) cells (Scavuzzo et al., 2018). However, the role of *AMOTL2* in endocrine specification and subtype differentiation in the small intestine remains to be characterized.

Next, we investigated transcriptional similarities between the identified cell types. Hierarchical clustering based on highly variable genes placed progenitor clusters (Proliferative progenitors, Progenitors, and NEUROG3 Progenitors) in a separate branch from hormone-producing cells (XMD-LK, EC early, and EC late) (Fig. 3G). Proliferative progenitors and Progenitors were more similar to each other than to NEUROG3 progenitors, which may reflect the still proliferative status of the former (Fig. S6A). On the other hand, EC clusters were closely related, suggesting that one population (EC early) is likely to be a precursor of the other (EC late). To define a step-wise lineage restriction of EEC in HIOs, we then performed partition-based graph abstraction (PAGA) analysis (Wolf et al., 2019). This analysis enables us to infer a complete lineage tree through graph-based topology analysis, measuring subgroup connectivity and providing a graph of possible differentiation paths (Fig. S6B). The thicker the line, the greater confidence in the linkage. Consistent with previous studies in rodents (Piccand et al., 2019) and humans (Burclaff et al., 2022; Wang et al., 2019), we found that

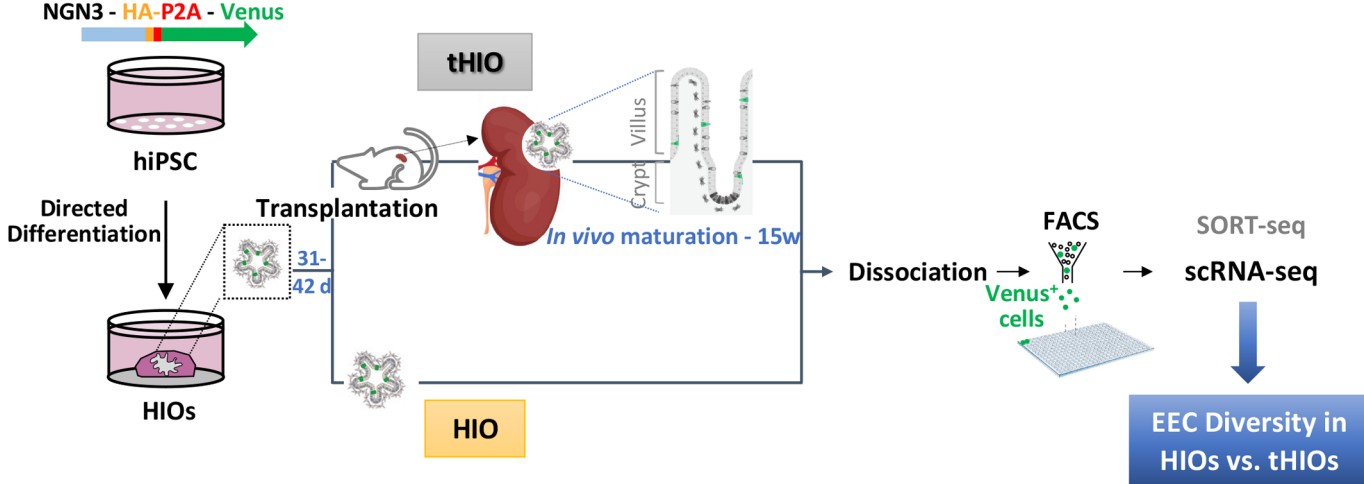

**Fig. 2. Worflow to generate, isolate and sequence hiPSC-derived human EECs.** HIOs were generated by directed differentiation of NGN3-HA-P2A-Venus hiPSCs, cells dissociated by enzymatic treatment after 31-42 days of culture and VENUS positive EECs sorted by FACS for scRNA-seq (~40 HIOs dissociated). Alternatively, HIOs were transplanted under the kidney capsule of immunodeficient mice for *in vivo* maturation during 15 weeks. tHIO (n=8) were then harvested and EECs sorted for scRNA-seq as described above.

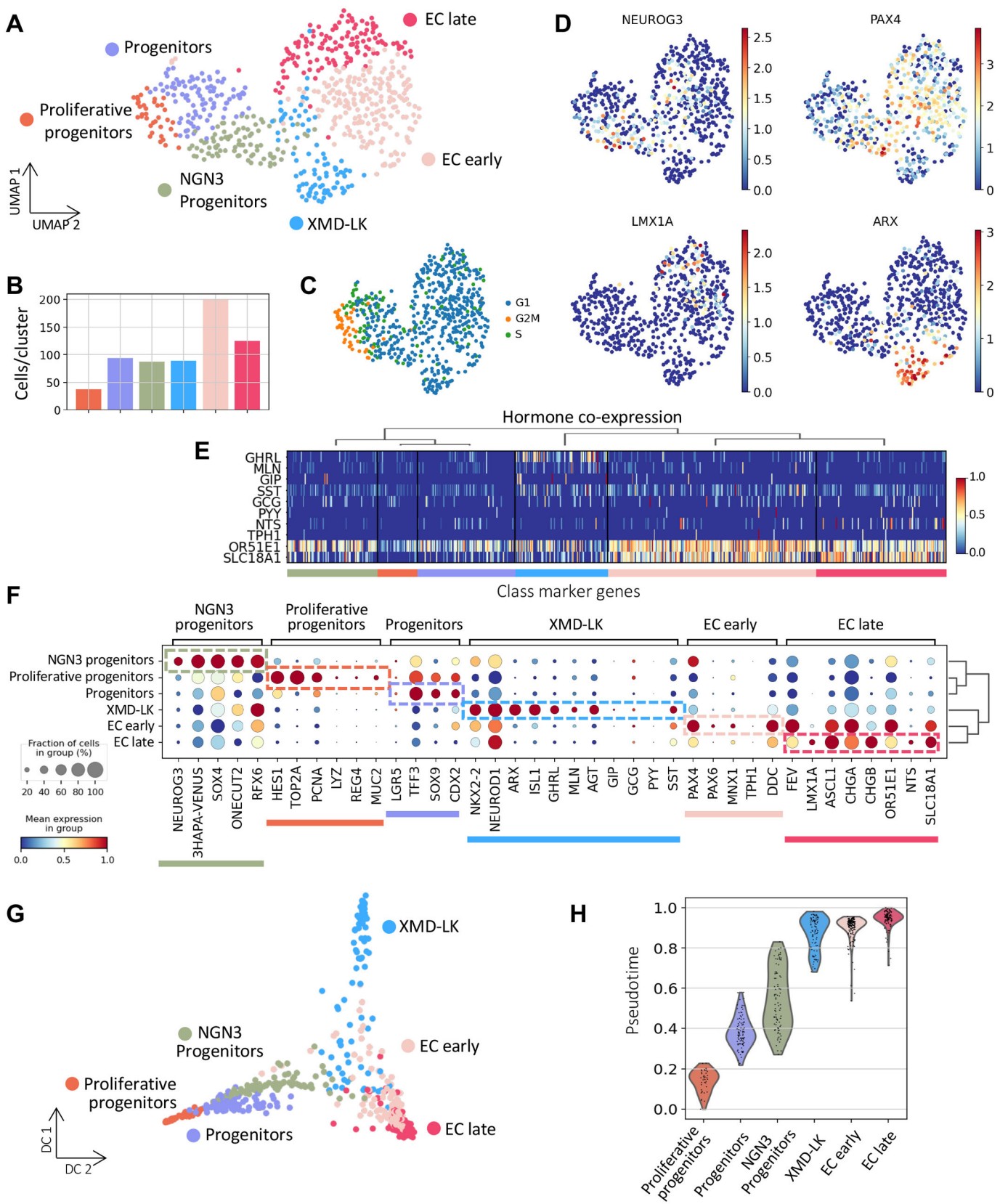

**Fig. 3.** See next page for legend.

NEUROG3 endocrine progenitors differentiate into two main branches, the EC and XMD-LK. We found strong connections of NEUROG3 progenitors to EC early and XMD-LK cells; and a weaker connection of these progenitors to EC late, while EC early and EC late cells were strongly connected. Pseudotemporal ordering of the cells using diffusion maps algorithm (Haghverdi et al., 2016)

**Fig. 3. Single cell transcriptome profiling of EEC from pluripotent stem cell-derived human intestinal organoids (HIOs) identifies two main differentiation branches.** (A) UMAP plot of 633 profiled cells from Neurog3-Venus derived HIOs (31 days). Colors highlight clustering into six main cell types. Including endocrine progenitors (Proliferative progenitors, Progenitors, and NGN3 Progenitors) and hormone-producing cells (XMD-LK, EC early, and EC late). (B) Bar plots showing the number of cells per cluster. (C) UMAP plot colored by the predicted cell cycle phase shows that a proportion of the progenitor population is still in the proliferative phase (G2M). (D) Gene expression profile of known marker genes for endocrine progenitors (NEUROG3), EC early (PAX4), EC late (LMX1A), and PE (ARX). Normalized expression values are shown. (E) Heatmap showing hormone co-expression along the different clusters. Cluster membership is indicated at the bottom. Gene expression values are normalized between zero and one. (F) Dotplot showing expression of known and uncharacterized cell type-specific gene sets of enteroendocrine cells. Color intensity indicates mean expression (normalized) in a cluster; dot size indicates the proportion of cells in a cluster expressing the gene. (G) Diffusion map representation of the profiled population. Colors highlight clustering into the six main cell types. It further confirms the differentiation of EECs in two different lineages. (H) Violin plot summarizing the pseudotime distribution per cluster. The mean of the pseudotime distributions progressively increased from progenitor clusters to hormone-producing cells.

showed that the EC and PE lineages bifurcate from NEUROG3 progenitors (Fig. 3G). Furthermore, pseudotime analysis suggested that EC early cells appear before EC late and XMD-LK cells, which start to differentiate around the same pseudotime (Fig. 3H). This supports the idea that endocrine progenitors differentiate into EC and PE cells. EC cells differentiate progressively, passing through different stable precursor populations.

Altogether, these results confirm that HIOs recapitulate EEC development in the human small intestine, occurring in two main lineages. However, although EC and XMD-LK resemble major EEC fate choices, they do not recapitulate the full diversity of PE cell subtypes. Indeed, we did not observe expression of the hormone CCK, and some other peptide hormones, like GIP, were very poorly expressed.

## Transplantation of human intestinal organoids enhances enteroendocrine cell diversity and maturation

Transplantation of HIOs into the kidney capsule of immunocompromised mice significantly enhances intestinal maturation (Watson et al., 2014) resulting in the formation of well-organized crypt-villus structures. Of note, some EEC cells, like CCK cells, differentiate only under transplantation conditions (Sinagoga et al., 2018). Therefore, we explored EECs diversity in tHIOs by transplanting NEUROG3-HA-P2A-Venus HIOs and analyzed cellular heterogeneity 15 weeks after transplantation (Fig. 2 and Fig. S7). After filtering, we profiled 560 cells including EPCAM+/VENUS+ and EPCAM+/VENUS− epithelial cells which comprised both non-EECs and EECs (Fig. 4A-B). EEC clusters, include NEUROG3 early progenitors (NEUROG3, SOX4, TOX, INSM1), NEUROG3 late progenitors (NEUROG3, RFX6, NKX2-2), two groups of peptidergic EECs, named according to the classical EEC nomenclature, i.e. XMD (GHRL, MLN, SST, ARX, ISL1) and LINKSD (PYY, GCG, CCK, NTS, GIP, SCT, SST, PAX6, ETV1), and two groups of EC cells, EC early (PAX4) and EC late (CHGA, TPH1, SLC18A1, LMX1A, FEV) (Fig. 4B). To better understand EEC diversity, we next focused on the endocrine populations (461 cells) (Fig. 4C-J), i.e., we computationally excluded Enterocytes, Tuft cells, Goblet cells. We identified seven cell groups using unsupervised clustering (Fig. 4C) and analysis of known marker genes. Figs S8 and S9 summarize the top 50 differentially expressed (DE) genes in each cluster. We noticed that among the progenitor populations

(Progenitors, NEUROG3 early and late progenitors), ∼10% were in the proliferative phase of the cell cycle (Fig. 4D). Additionally, we observed that PAX4 was enriched in early EC cells. Conversely, LMX1A and ARX were in more mature cells and seemed mutually exclusive, defining the prototypical EC and PE branches, respectively (EC late versus XMD, LINKSD) (Fig. 4E). Similar to what we obtained in HIOs and consistent with previous reports on adult human EECs (Beumer et al., 2018), we found a subpopulation, the XMD cluster, defined by GHRL and MLN expression (Fig. 4F). Importantly, we confirmed that transplantation improved hormone cell diversity and maturation. Indeed, we obtained cells co-expressing various combinations of hormone genes i.e. PYY, GCG, CCK, NTS, GIP, SCT, and SST (Fig. 4F), which were not previously enhanced in the HIO system. Transcriptional similarity analysis based on highly variable genes and hierarchical clustering placed progenitors and mature hormone-producing cells in different branches (Fig. S6C). Interestingly, EC early cells were highly correlated with NEUROG3 late progenitors, while hierarchical clustering placed them closer to EC late and LINKSD cells. To understand the possible differentiation paths, we performed graph-based topology analysis (PAGA). As expected, there was a progressive transition of the progenitor populations, with a strong connection of NEUROG3 late progenitors to the hormone-producing cells. LINKSD cells had a strong link with XMD, as expected, and strikingly also with EC cells (early and late), suggesting that some PE cells share differentiation pathways with EC cells (Fig. S6D). This is the case in the mouse, where Pax4 knockout strongly decreased the expression of Gip, Nts and Tph1 (5-HT cells) in the small intestine (Beucher et al., 2012). Further pseudotemporal ordering of the cells using diffusion maps showed that PE cells, instead of following a progressive differentiation, bifurcate into two separate branches: XMD, and LINKSD, which are also distinct from the EC branch. Notably, EC cells showed a similar progression as before transplantation (Fig. 4G). Pseudotime analysis suggested that XMD and LINKSD arise at similar times; and as expected, EC early cells start to differentiate before EC late cells (Fig. 4H).

We next used the available human intestinal atlas (Elmentaite et al., 2021) to compare the EEC types obtained in tHIOs (Fig. 4I,J). The Elmentaite et al. study profiled EECs derived from the human intestinal epithelium at different ages (first and second fetal trimesters, pediatric and adult) and from different regions (proximal and distal small intestine, colon and rectum). Joint embedding of the two single cell RNA-seq data sets revealed that the transcriptomic signature of EECs in tHIOs resemble EECs isolated from primary human intestine. tHIOs cells segregate predominantly with cells derived from the human fetal proximal small intestine, as expected in our model. In particular, no tHIOs cells were found in the 'EC cells (NPW+)' cluster, which essentially corresponds to adult large intestine cells. Also, very few or no tHIOs cells segregate into the 'L cells (PYY+)' and 'N cells (NTS+)' clusters, respectively, two cell subtypes that are more abundant and characteristic of the distal intestine. Although these subtypes were clearly identified in tHIOs, cell numbers remain low. Interestingly, the other tHIO clusters fit into the atlas clusters with closely related signatures. This demonstrates proper cluster definition and confirms that the tHIO model faithfully recapitulates EEC differentiation in the developing proximal small intestine.

Taken together, these results indicate that EECs generated in tHIOs resemble primary human small intestinal EECs, and that transplantation enhances EEC diversity, particularly for PE cells. Furthermore, we demonstrated that human EECs differentiate along three distinct pathways, differing from previous reports in adult mice, which identified only two differentiation branches (Gehart et al., 2019; Piccand et al., 2019).

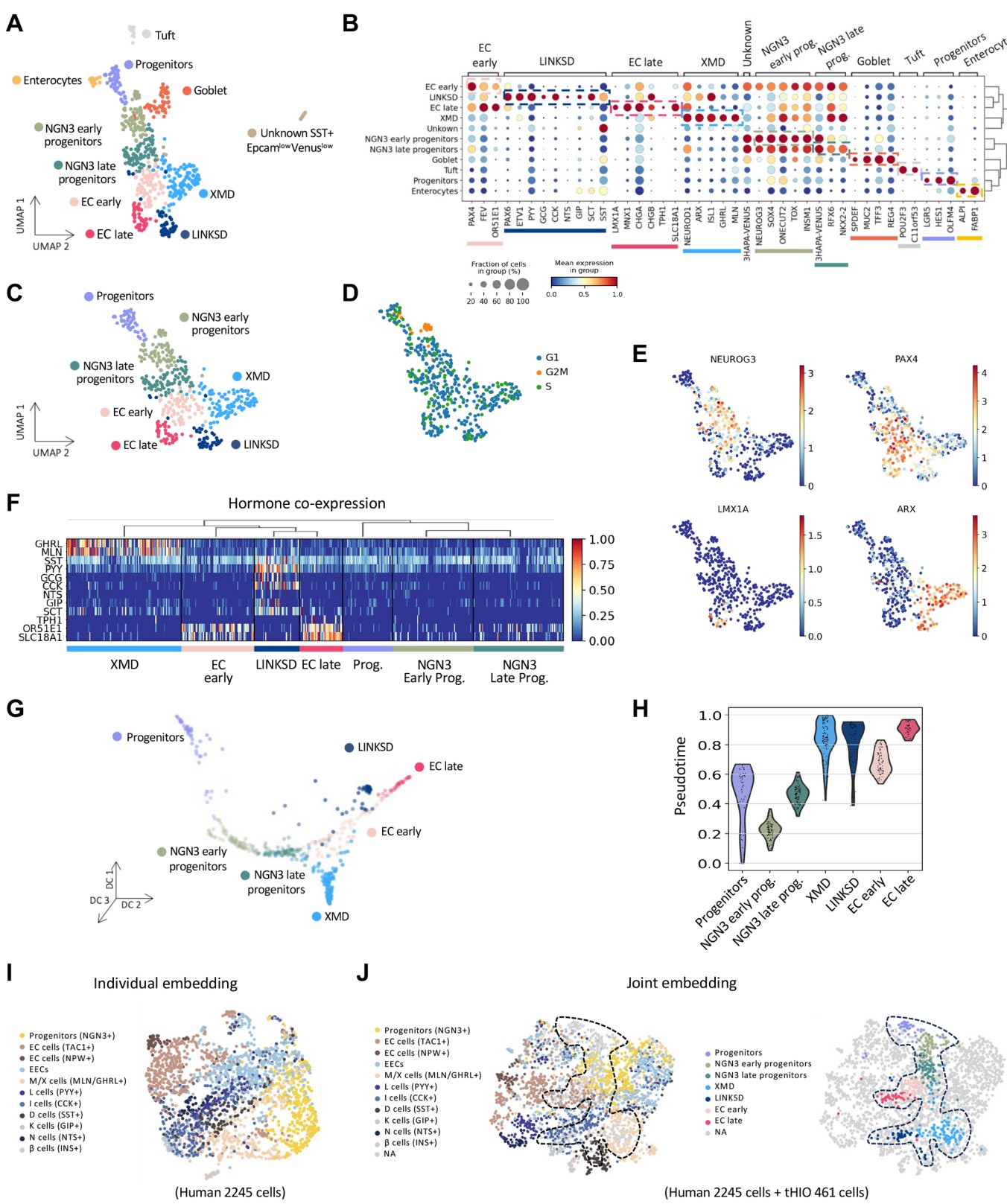

**Fig. 4.** See next page for legend.

## Comparing HIOs versus tHIOs reveals molecular signs of EEC maturation

To further investigate the molecular mechanisms of EEC determination before and after transplantation, we performed pairwise differential expression tests for EEC Progenitors and Hormone-cells between HIOs and tHIOs (Fig. 5A and B). Progenitor populations in HIOs had higher expression of pluripotency and early endoderm markers such as *SOX17* and *CXCR4*. Conversely, progenitors in tHIOs expressed high levels of *GATA4* or EEC precursor genes, such as OLFM4, which, although a

**Fig. 4. Transplanted human intestinal organoids enhance EEC diversity and maturation.** (A) UMAP plot of 560 profiled cells from Neurog3-Venus derived tHIOs (*n*=8). Colors highlight clustering into eleven main cell types including non-EECs and EECs (Progenitors, NGN3 early, NGN3 late, EC early, EC late, XMD, and LINKSD). (B) Dot plot showing expression cell type-specific gene sets. Color intensity indicates mean expression (normalized) in a cluster; dot size indicates the proportion of cells in a cluster expressing the gene. (C) UMAP plot focused on 461 EECs from tHIO. D. UMAP plot colored by the predicted cell cycle phase shows that a small fraction of the NGN3 early progenitors' population is still in the proliferative phase (G2M). (E) Gene expression profile of known marker genes for NGN3 progenitors (NEUROG3), EC early (PAX4), EC late (LMX1A), and cells expressing peptide-like hormones (ARX). Normalized expression values are shown. (F) Heatmap showing hormone co-expression along the different clusters. Cluster membership is indicated at the bottom. Gene expression values are normalized between zero and one. (G) Diffusion maps representation of the EEC population in tHIOs. Colors highlight clustering into the seven main cell types. It further confirms the differentiation of EECs in three different lineages. (H) Violin plot summarizing the pseudotime distribution per cluster. The mean of the pseudotime distributions progressively increased from progenitor clusters to hormone-producing cells. (I,J) Integration of tHIO EEC transcriptomic profiles with human EEC. (I) UMAP Individual embedding of human primary intestinal EECs (Elmentaite et al., 2021). (J) UMAP combined embedding of primary and tHIO human EECs.

stem cell marker, is more abundant in the transit amplifying zone where progenitors reside (Burclaff et al., 2022); *NEUROG3*, *NKX2-2*, *ARX*, *PDX1*, *MAFA*, and *REG4* were also enriched in the tHIO progenitor population. On the other hand, Hormone-cells in HIOs presented a high expression of EC markers (*CHGA*, *SLC18A1*, *OR51E1*) as well as of the bHLH transcription factor *ASCL1* (Fig. 5A and B); surprisingly, insulin (*INS*) had a high fold change in this population. In this line, Egozi et al. (2021) found *INS* expression in the fetal small intestine (Egozi et al., 2021), which may suggest that HIOs resemble more immature, developing tissue. In contrast, peptide-like hormone genes (*GIP*, *CCK*, *GHRL*, *MLN*, *SST*, *PYY*) and enzymes involved in the peptide hormone biosynthesis (*CPE*) (Bär et al., 2014) were enriched in tHIOs. Moreover, potassium-channel-related genes (e.g. KCNK17) were overexpressed in tHIOs. These genes have been shown to be essential for maintaining homeostasis in the gastrointestinal epithelia (Heitzmann and Warth, 2008). Gene set enrichment analysis on the hormone cells (Fig. 5C) showed that genes upregulated in HIOs were more involved in glucose metabolism. In contrast, those in tHIOs exhibited intestinal epithelial cell differentiation, regulation of peptide hormone secretion, and calcium ion import hallmarks. Notably, previous studies have shown that calcium signaling induces hormone secretion (Goldspink et al., 2018), suggesting that Hormone cells in tHIOs might be functional. This result further supports the previous conclusion from differential gene expression analysis, indicating that tHIOs EECs exhibit maturation and functional features. It is worth noting however, that Sinagoga et al. (2018) demonstrated that HIOs, even prior to transplantation, were already functional by hormone secretion in response to luminal glucose (Sinagoga et al., 2018).

In summary, the differential expression analysis of progenitor and hormone-expressing cells in HIOs and tHIOs confirms that tHIOs give rise to more mature EECs, while HIOs remain immature and retain early endoderm signatures. Therefore, we conclude that although human intestinal organoids are capable of differentiating EECs, it is only transplantation and the molecular mechanisms that it enables, that enhance EEC diversity, especially for the peptidergic lineage. However, alongside *in vivo* maturation, some of the

differences in gene expression observed between HIO and tHIO EECs may also, at least in part, arise from variations in organoid culture conditions.

## Cell surface markers of human EECs

To identify which pathway that could modulate the biological activity of human EECs we examined the expression of cell surface receptors. Interestingly, we found that *EPHA4*, a membrane receptor tyrosine kinase (RTK) of the EPH family, involved in cell migration and adhesion (Poliakov et al., 2004), was differentially expressed in the XMD cluster in tHIOs (Fig. 6B). EPHA4 was also differentially expressed in the XM cluster in HIOs and in the X cluster in the mouse embryonic intestine (not shown) supporting that EPHA4 is an early marker for the ghrelin-expressing subset. The intestinal function of EPHA4 is unknown, but it has been shown to inhibit glucagon secretion from pancreatic α cells (Hutchens and Piston, 2015), suggesting it could regulate Ghrelin secretion from EECs. Two other atypical receptors, part of the integrin family responsible for cell-extracellular matrix (ECM) interactions are differentially expressed in the XMD cluster in tHIOs (Fig. 6C,D) and HIOs but not in the mouse EECs. These are the β8 subunit (ITGB8) which forms the αvβ8 integrin and the α7 subunit (ITGA7) which forms the α7β1 integrin. Integrin αvβ8 is a receptor for ECM-bound forms of latent transforming growth factor β (TGFβ) proteins and promotes the activation of TGFβ signaling pathways (McCarty, 2020). Integrin subunit α1 (ITGA1) (Fig. 6E), also known as CD49a, which has been used to purify β-cells from SC-islets (Veres et al., 2019), as well as Transmembrane protein 27 (TMEM27) transcripts (Fig. 6F) are enriched in the LINKSD subpopulation. Other cell surface receptors are enriched in EC cells, i.e. OR51E1 (Fig. 6G) or SLC18A1 (not shown). These findings highlight the importance of cell-cell and cell-ECM/microenvironment interactions in the differentiation or function of human EECs and could be leveraged to purify EEC subtypes.

## Differentiation trajectories of EEC populations in tHIOs

We then sought to identify the distinct transcriptional interactions leading to the differentiation of all hormone cell populations in tHIOs. To that end, we used FateCompass (Jiménez et al., 2023) to infer lineage-specific regulators of EECs. Briefly, FateCompass implements a three-step analysis process: first, it reconstructs differentiation trajectories to capture dynamic cellular transitions; second, it learns TF activities using a linear model of gene regulation; and third, it identifies key regulators of cell subtype specification by integrating dynamic changes in TF activity over differentiation trajectories and expression correlations.

To infer differentiation trajectories, FateCompass performs stochastic simulations of cell state transitions using a probabilistic framework biasing the transitions using a potential energy gradient from progenitor cells to each terminal fate (EC late, XMD, and LINKSD) (Fig. 7A,B). In Fig. 7C, we present representative differentiation trajectories for each of the final cell types. Finally, based on the stochastic simulations, we estimated fate probabilities of each cell to become a specific cell type (Fig. 7D). We observed that the fate probabilities for the EC and XMD cells were high since the early stages of differentiation. In contrast, the probability of becoming a LINKSD cell was lower and increased only at late stages. Alternatively, intermediate states might be rare or transient and thus more difficult to capture for the LINKSD fate. This observation suggests that EC and XMD cells are transcriptionally closer to EEC progenitors. This is consistent with our previous observations of EC cells and Ghrelin (X)/Motilin (M) cells

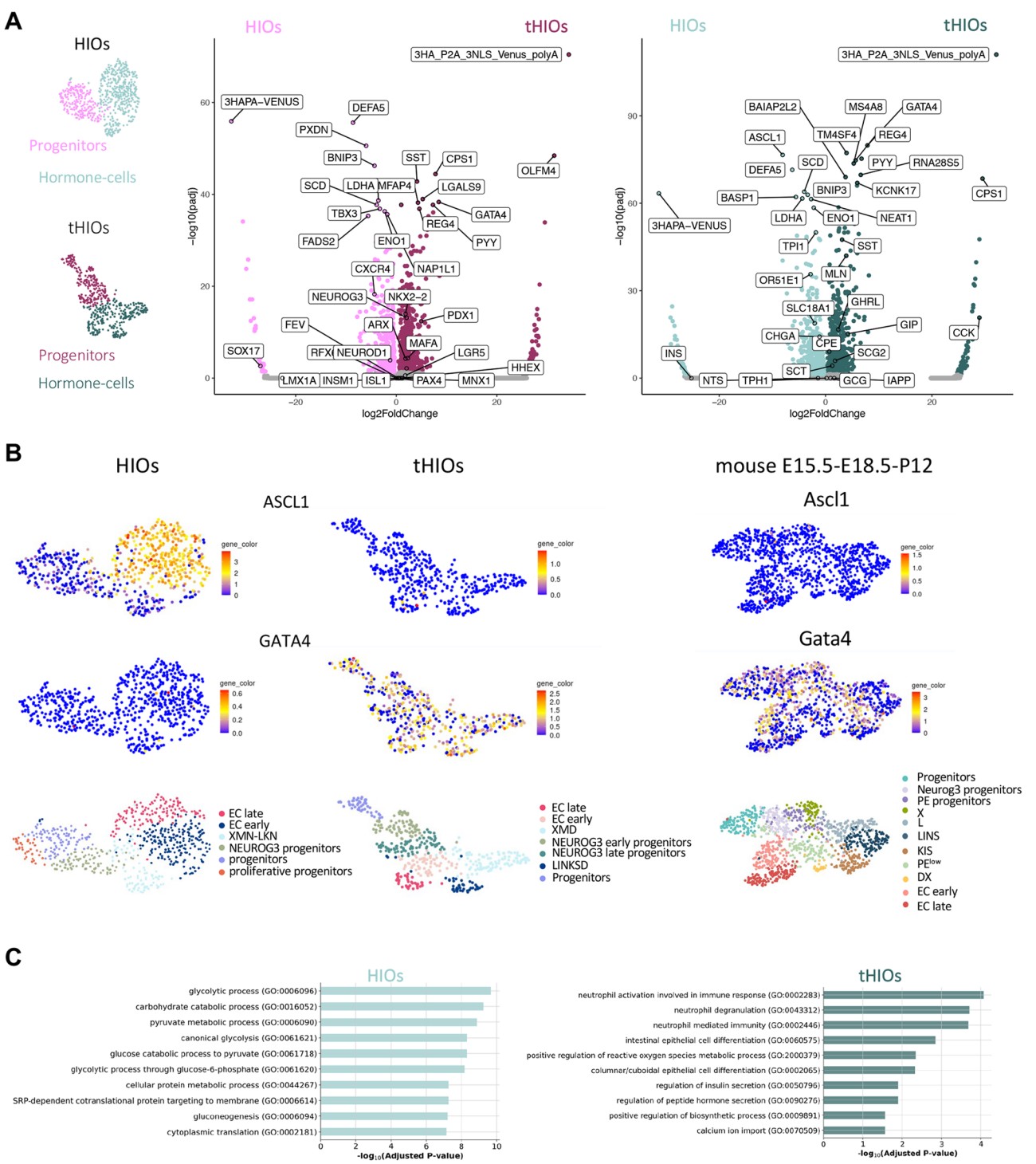

**Fig. 5. Comparing HIOs vs tHIOs reveals signs of EEC maturation.** (A) Differential gene expression analysis of progenitor populations and hormone-cells between HIOs and tHIOs. Progenitors and hormone cells in tHIOs differentially expressed mature EEC cell markers. (B) UMAPs illustrating the expression of ASCL1 and GATA4 transcription factors mRNAs in EECs from HIOs or tHIOs (left and middle panels) or mouse developing small intestine (right panels). (C) Gene set enrichment analysis based on GO terms of Biological Process with differentially expressed genes on hormone-cells from HIOs and tHIOs. While HIOs present more glucose metabolism functions, tHIOs have terms associated with possible functionality of EEC (regulation of insulin and peptide hormone secretion and calcium ion import).

differentiating even in a less mature setting, i.e., in HIOs before transplantation. Next, we examined the expression profiles of hormone products of each terminal cell type across differentiation trajectories (Fig. 7E). We reasoned that the dynamic profile of each hormone should increase over the trajectories of the cell types

it defines. As expected, serotonin-related markers (*TPH1*, *OR51E1*, and *SLC18A1*) had an increasing profile only in the EC trajectories. The hunger hormone *GHRL* and the human specific motility hormone *MLN* increased over the XMD trajectories. *SST* presented a high basal level in the early stages of the three differentiation

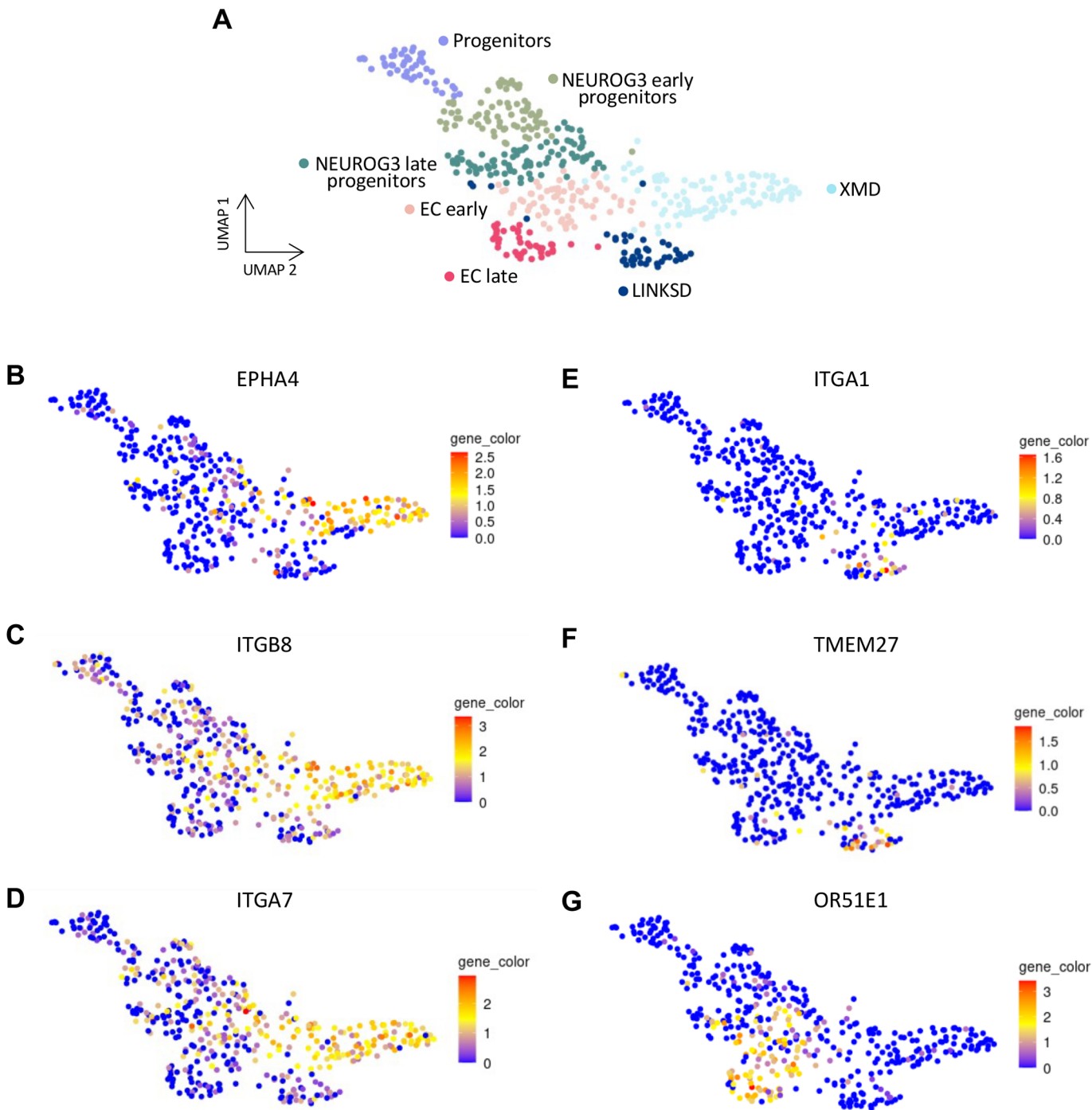

**Fig. 6. Novel cell surface markers of human EECs.** (A) UMAP plot showing clusters of the main cell types identified in tHIOs. (B-G) Gene expression profile of receptor genes, EPHA4 (B), ITGB8 (C), ITGA7 (D), ITGA1 (E), TMEM27 (F) and OR51E1 (G) in tHIOs, showing specific markers of the XMD (B-D), LINKSD (E-F), and EC (G) clusters, respectively. Normalized expression values are shown.

pathways; thereafter, its profile increased in XMD and LINKSD, while decreasing in EC. On the other hand, except for *SCT*, which increased over both LINKSD and XMD trajectories, L cells hormones (*PYY* and *GCG*), *CCK*, *NTS*, and *GIP*, presented an increasing profile only over LINKSD trajectories, indicating that these hormones are highly specific to this lineage. In summary, the dynamic profiles inferred by FateCompass accurately described the expression progression over the EEC differentiation trajectories.

**Transcription factor activity dynamics during human EEC differentiation**

Next, we used FateCompass to infer TF activities dynamically. The linear model of gene regulation implemented in FateCompass estimates the activity of TFs in single cells by statistically evaluating whether their target genes are expressed in a coordinated manner relative to the average gene expression, while also disentangling the influence of multiple TFs on shared target genes. In Fig. 8, we showed the expression and activity distribution in the UMAP

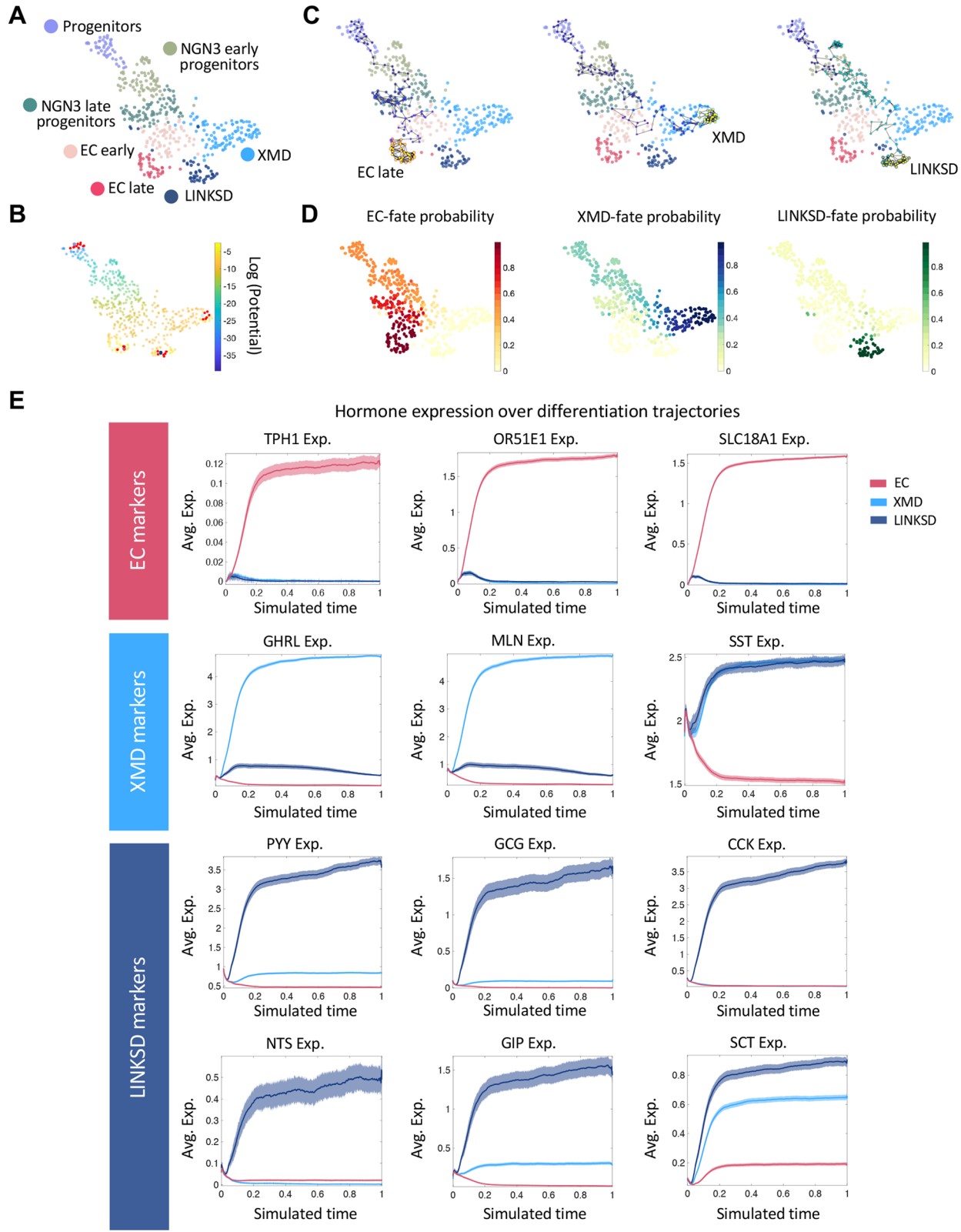

**Fig. 7. Differentiation trajectories and EEC subtypes specification using tHIOs.** (A) UMAP plot of 461 EECs from tHIOs. (B) UMAP plot colored according to the potential energy, the gradient goes from Progenitor cells (source) to the mature hormone-producing cell types: EC late, XMD, and LINKSD (sinks). Red dots represent the possible neighborhood a cell can explore when modeled using a Markov chain. (C) Stochastic differentiation trajectories start in Progenitor cells and end in EC, XMD, and LINKSD fates after 2000 iterations. FateCompass delineates the cell subtype specification process by biasing transition probabilities from Progenitor cells toward hormone-producing cells. (D) Fate probabilities are calculated as the absorption probabilities for the EC-, XMD- and LINKSD-fate. The likelihood of ending up in EC and XMD fates is high since the early stages, while the one of LINKSD only increased at late stages. (E) Average gene expression of known lineage-specific markers over EC, XMD, and LINKSD stochastic differentiation trajectories. The dynamic profile of each cell type marker increased over the trajectories of the lineage they defined.

embedding, the expression and activity profile over differentiation trajectories, and the cross-correlation between each lineage's expression and activity profiles for some known regulators. Briefly, we computed Pearson correlation at different time lags considering that the activity of a TF could be delayed with respect to the expression of its mRNA or vice versa. A strong positive correlation between the TF activity and the TF mRNA expression is an indication that the TF behaves as an activator; otherwise, a strong negative correlation suggests a possible repressor role. We first focused on *ARX* (Fig. 8A), since previous studies showed that *Arx*-deficient mice display almost complete loss of *GLP1*, *GIP*, *CCK*, and *NTS*, with a concomitant increase of *SST* cells (Beucher et al., 2012; Du et al., 2012; Terry et al., 2018). Also, lineage-tracing of EEC progenitors and their descendants using a *Neurog3* reporter mouse at the single cell level showed that *Arx* is an essential marker of PE precursors and GHRL cells (Gehart et al., 2019; Piccand et al., 2019). Arx expression was shown to behave similarly in mouse *in vivo* and human enteroids *in vitro,* with expression in most PE cells (Beumer et al., 2020). In agreement with these reports, we observed an increasing *ARX* expression profile only in XMD- and LINKSD-cells trajectories. The activity was primarily high in XMD-cells, where the correlation analysis indicates an activator role. TF activity was negative in LINKSD cells, suggesting different regulatory mechanisms for *GLP1*-, *GIP*-, *CCK*-, and *NTS*-expressing cells in humans. Thus, while *Arx* is essentially known as a repressor, it could also act as an activator depending on the cellular context. Next, we looked at *PAX4*, since it is known that Arx and Pax4 have opposing functions in the specification of

endocrine cells in the pancreas and the intestine, respectively (Beucher et al., 2012; Collombat et al., 2003; Larsson et al., 1998) . In the mouse gut, serotonin-producing cells are exclusively Pax4-dependent (Beucher et al., 2012). As expected, *PAX4* expression and activity increased over EC trajectories (Fig. 8B). The correlation between the two profiles is strongly positive, suggesting similar mechanisms in mice and humans for *PAX4*; this is in line with Beumer et al. (2020) observations on EEC transcriptomes in human-derived enteroids (Beumer et al., 2020; Lin et al., 2023). Furthermore, our results suggest that PAX4 acts as a transcriptional activator to control EC features. The LIM homeobox transcription factor 1 alpha (Lmx1a) has been shown to regulate *Chga* and *Tph1* in the mouse (Gross et al., 2016), and is a well-known marker of EC fate (Piccand et al., 2019). Interestingly, EC cells exhibit high *LMX1A* expression correlating with gene accessibility (Hickey et al., 2023). Here, we observed increasing *LMX1A* expression and activity over EC trajectories, showing a positive correlation, suggesting an activator role (Fig. 8C) in the GRNs underlying EC-fate determination.

### Transcriptional interactions underlying enteroendocrine subtype specification in tHIOS

To identify lineage-specific regulators dynamically, we leverage the differential motif activity analysis of FateCompass. We defined a differential motif activity analysis based on the following criteria: (i) motifs with the high positive z-score, i.e., motifs that significantly varied across cells compared with their estimated errors; (ii) motifs with high activity variability across the

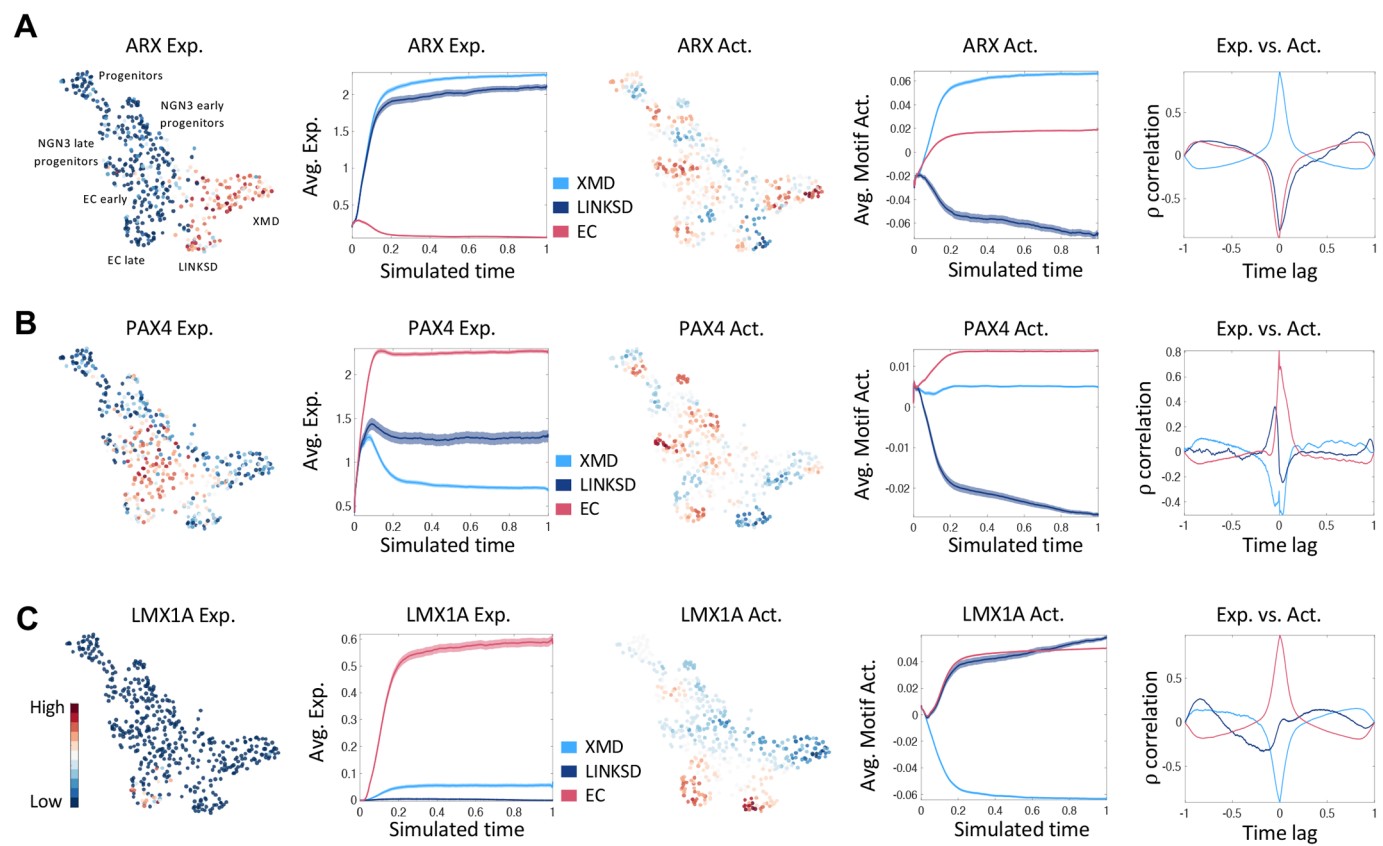

**Fig. 8. Activity profile characterization in tHIOs of known EEC regulators in the mouse.** For each TF we present the gene expression profile in UMAP plots (normalized expression values are shown), the average gene expression profiles over stochastic differentiation trajectories, the activity profile in UMAP plots, the average activity profile over stochastic differentiation trajectories, and the Pearson correlation between the dynamic expression and activity at different time lags for ARX (A), PAX4 (B), LMX1A (C). Exp., expression; Act., activity.

lineage-specific differentiation trajectory. Finally, (iii) motifs with a high temporal correlation between its activity and mRNA expression within a specific window of time lags (Jiménez et al., 2023). We identified 133 differentially active TFs (Fig. 9, Table S1), 13 for the three EEC lineages (Fig. 9A), 5 for both EC and XMD (Fig. 9B), 15 for EC and LINKSD (Fig. 9C), 9 for XMD and LINKSD (Fig. 9D), 34 were EC-specific (Fig. 9E), 42 were XMD-specific (Fig. 9F), and 15 were LINKSD-specific (Fig. 9G). CDX1 and RBPJ were predicted for all the hormone cells (Fig. 9A), and their activity was high early on during the differentiation trajectories; these factors have been reported to play a role in defining patterns of proliferation and differentiation along the crypt–villus axis (Silberg et al., 2000) and as a downstream effector of the Notch signaling pathway (Gehart and Clevers, 2019), respectively. The above suggests that factors common to the three EEC lineages act during early fate choices. Next, we investigate the factors common for two fates simultaneously. PAX6 and NKX2-2 were predicted as EC- and LINKSD-specific (Fig. 9C and Fig. S10A-B). In the case of PAX6, the activity profile is milder at the beginning of the trajectories; then, it goes up and stays at a high level only for LINKSD cells. Studies in the mouse (Larsson et al., 1998) and human (Fujita et al., 2008) have shown that PAX6 is important for GIP-producing cells. Interestingly, previous studies in the mouse intestine proposed that Nkx2-2 acts upstream Pax6 (Desai et al., 2008), strengthening the fact that they are found together in a regulatory program. Moreover, Nkx2-2 deletion in Neurog3 positive EEC progenitor cells in the embryo and adult mouse results in the loss of most EEC types and an increase in the ghrelin cell population within the duodenum (Gross et al., 2016). Regarding the factors identified for XMD and LINKSD cells, we found ISL1 (Fig. 9D). Gene ablation of Isl1 in the mouse intestine results in loss of GLP-1, GIP, CCK, and SST-expressing cells and an increase in 5-HT (serotonin)-producing cells (Terry et al., 2014). Together, an inspection of the factors predicted for two lineages simultaneously shows high similarities with functional studies reported in mice, indicating that some of the shared programs between different EEC subtypes might be conserved. Finally, we focused on FateCompass predictions for each particular lineage. As expected, LMX1A was identified as EC-specific. Interestingly, we predicted CDX2 also as EC-specific (Fig. 9E; Fig. S10C). In the pancreas, during *in vitro* differentiation towards β-like cells, there is a cell type that exhibits EC-like features that we predicted and experimentally validated CDX2 to be a driver of this population (Jiménez et al., 2023). Accordingly, others showed that CDX2 regulates serotonin-positive cells which have been proposed to mark a pre-β−cell population in the human fetal pancreas (Zhu et al., 2023). Together, these findings suggest that CDX2 might have different roles during intestinal differentiation, early on for the patterning of the intestinal epithelium (Gao et al., 2009) and later on for EC cell subtype specification. ARX was among the XMD-specific factors (Figs 8A and 9F), pinpointing a function only in a subset of PE cells; similar results in the mouse, based only on expression, restrict ARX to the X cells (Gehart et al., 2019). Vitamin D receptor (VDR) was also predicted as XMD-specific (Fig. 9F; Fig. S10D). Previous studies showed that vitamin D3 induces intestinal epithelial cell differentiation in mouse intestinal organoids (Sittipo et al., 2021). Furthermore, studies in patients with type 2 diabetes suggested that vitamin D intake increases circulating ghrelin (Hajimohammadi et al., 2017); whether VDR has a role in the differentiation of ghrelin-cells (X) remains to be tested. PDX1 was identified as LINKSD-specific (Fig. 9G and Fig. S10E). Fujita et al. (2008) reported that concomitant expression of PDX1 and PAX6 is

essential for L cells that co-express GIP (Fujita et al., 2008), which is consistent with FateCompass predictions. SNAI1 was predicted active during the initial phase of LINKSD cel differentiation (Fig. 9G), and has previously been shown to regulate EEC differentiation in mice (Horvay et al., 2015). PROX1 was also LINKSD-specific (Fig. 9G and Fig. S10F); notably, PROX1-positive EECs have been reported to possess intestinal stem cell activity during both homeostasis and injury-induced regeneration (Yan et al., 2017). Furthermore, FateCompass identified PROX1 as a novel driver of glucagon-producing cells during mouse pancreas development (Jiménez et al., 2023) and Prox1 has also been found as enriched in mouse intestinal L-cells (Habib et al., 2012). Additional experimental validation will be important for elucidating shared regulatory mechanisms during glucagon-cell subtype specification in both the pancreas and intestine.

In summary, our integrative approach accurately predicted well-known and novel potential drivers of EEC cell subtypes. Using stochastic gene expression trajectories, we recapitulated cell subtype-specific trends and accurately described the dynamic progression of the markers for each hormone-cell type. Also, considering interactions between transcription factors and promoters dynamically, we predicted lineage-specific interactions. TFs predicted for all the hormone cell subtypes had functions at the early stages of EEC differentiation. Some of the interactions predicted for two lineages simultaneously indicated a conserved function with other animal models. Finally, putative drivers identified for EC cells and GCG-cells were similar for the pancreas (Jiménez et al., 2023), pointing to the similar regulatory mechanisms in these two tissues and reassuring the potential of the novel predictions.

## DISCUSSION

To elucidate the cellular mechanisms of EEC differentiation during development, we first mapped their differentiation trajectories at two key embryonic stages when EECs arise and develop in the mouse. Our findings reveal that, in the developing mouse intestine, EECs follow two primary differentiation pathways – the EC (Enterochromafin cells) and PE (Peptidergic EEC) pathways – similar to those observed in the adult intestine. Notably, these pathways are established at the earliest stage (E15.5), occurring alongside villi elongation but before the formation of crypts and the influence of environmental signals (such as dietary stimuli and microbiota). Additionally, all hormone-expressing clusters previously identified in adults are present at this early stage. This model, therefore, demonstrates for the first time, *in vivo*, that enteroendocrine specification is determined prior to the completion of intestinal morphogenesis. The study also confirms that most EEC subtypes co-express multiple hormones from the outset of development, and that the hormonal repertoire evolves as tissue maturation progresses. Specifically, as differentiation progresses within the EC lineage, the expression of early markers such as Tac1 in 'EC early' cells transitions to the expression of mature cell markers like Tph1 in the 'EC' subset, consistent with observations in the adult small intestine (Beumer et al., 2018).

Next, we characterized human enteroendocrine cell (EEC) differentiation, *in vitro*, using pluripotent stem cell-derived human intestinal organoids (HIOs) as models for human EEC development. This was achieved by leveraging single-cell transcriptomic data and FateCompass predictions. We profiled a substantial population of enteroendocrine cells (EECs) from HIOs and tHIOs without having to overexpress NEUROG3 to enhance EEC differentiation (Beumer et al., 2020; Sinagoga et al., 2018). This approach allowed us to investigate EEC heterogeneity and the mechanisms of differentiation from endocrine progenitors to hormone-producing

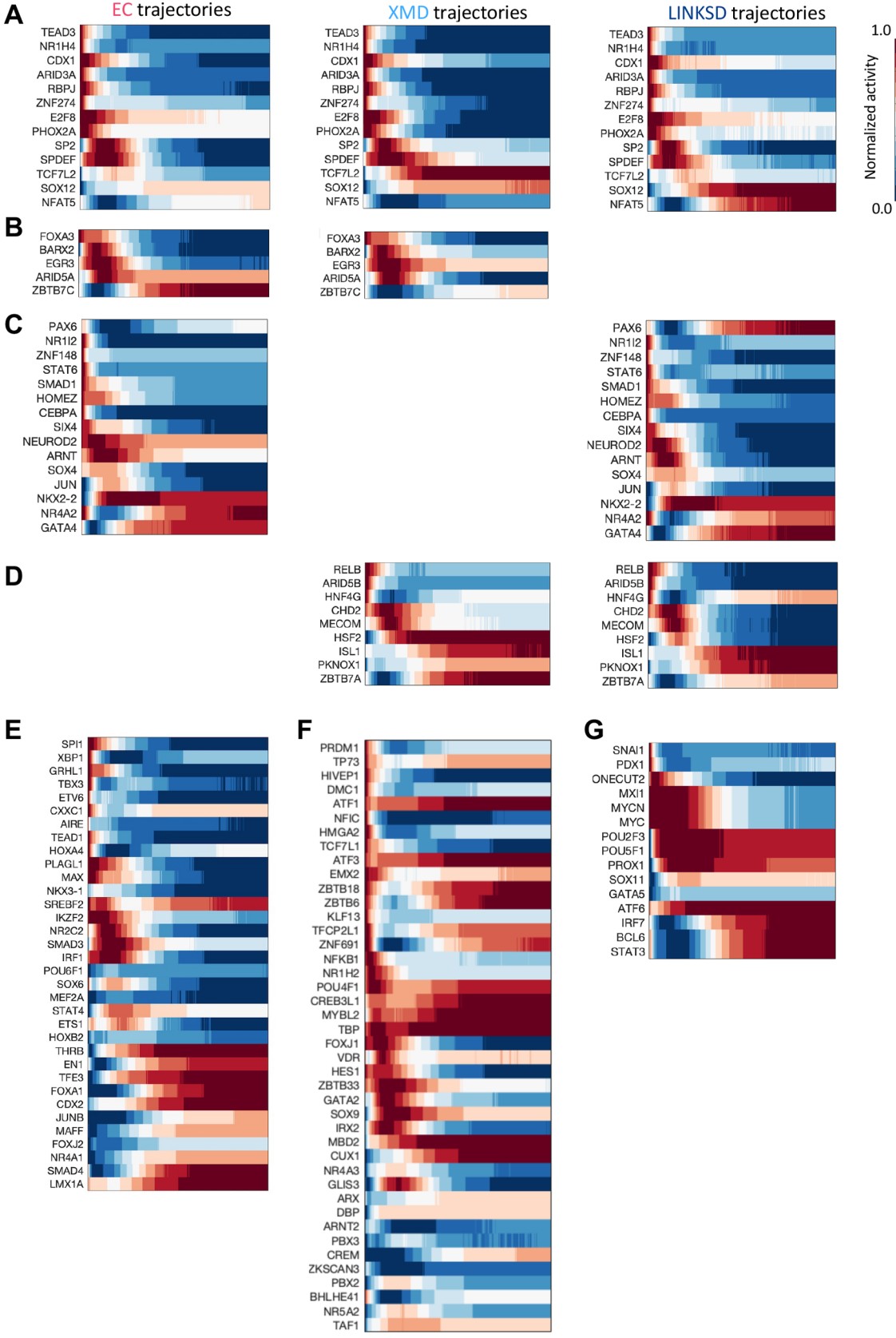

**Fig. 9. Differential motif activity analysis during human EEC differentiation.** Heatmaps showing the average motif activity over stochastic trajectories. (A) Factors predicted as drivers of the three lineages: EC, XMD, and LINKSD. (B) Factors predicted as drivers of both EC and XMD fates. (C) Factors predicted as drivers of both EC and LINKSD fates. (D) Factors predicted as drivers of both XMD and LINKSD fates. (E) Factors predicted as EC-specific sorted according to which was active first in the EC trajectories. (F) Factors predicted as XMD-specific sorted according to which was active first in the XMD trajectories. (G) Factors predicted as LINKSD-specific sorted according to which was active first in the LINKSD trajectories.

cells while avoiding the potential adverse effects on GRNs associated with NEUROG3 overexpression. HIOs largely mimic human EEC differentiation. Indeed, we observed that the genes encoding the human-specific hormone MLN co-expressed with the hunger hormone GHRL; this is consistent with previous publications reporting a transcriptomic atlas of EEC using enteroids (Beumer et al., 2018) and more recently human intestine (Elmentaite et al., 2021). EEC differentiation in HIOs follows a similar pathway as in the adult mouse intestine (Gehart et al., 2019; Piccand et al., 2019) suggesting conserved regulatory mechanisms. Furthermore, we confirmed that transplantation enhances intestinal epithelial maturation and promotes EEC diversity (Sinagoga et al., 2018; Singh et al., 2023; Yu et al., 2021). Unlike in adult mice, where all peptide hormone-producing cells are assigned to the same differentiation branch (Gehart et al., 2019; Piccand et al., 2019), we observed a new differentiation branch for peptidergic cells expressing *PYY*, *GCG*, *CCK*, *NTS*, *GIP*, *SCT*, and *SST*. Based on cluster connectivity analysis and trajectory inference, we hypothesize that this branch shares some regulatory mechanisms with the XMD and EC fates.

By comparing the signatures of HIOs and tHIOs, we identified hallmarks of epithelial maturation and molecular signatures that contribute to the increased diversity of hormone-producing cells in tHIOs. For example, transplantation of HIOs was associated with an increased expression of the stem cell marker OLFM4 (van der Flier et al., 2009) and REG4, with REG4+ secretory cells known to provide a niche function and promoting organoid formation in the mouse (Sasaki et al., 2016). Interestingly we found that the transcription factor ASCL1 is upregulated in HIOs within the population of immature EC-cells. In tHIOs, however, ASCL1 is only expressed in few EC cells. This finding is consistent with the recent research showing that ASCL1 restrains EEC differentiation and ensures accurate cell diversity in a 2D human intestinal stem cell culture where EEC differentiation was induced by transient NEUROG3 activation (Singh et al., 2024). Our finding thus suggests that ASCL1 may play an early role in human intestinal development specifically in EC cell differentiation. Strikingly, we observed a distinct and widespread expression of GATA4 in EECs from tHIOs, whereas only a few EE cells expressed this transcription factor in HIOs. This contrasts with the mouse model, where GATA4 is broadly expressed in embryonic EEC but reportedly absent in the EEC lineage in adult mice, GATA4 playing a role in the maintenance of the jejunal-ileal identity (Bosse et al., 2006). Interestingly, inactivation of GATA4 in the mouse jejunum led to a partial redistribution of the enteroendocrine population toward an ileal-like composition, with a decrease in CCK and an increase in PYY mRNA levels (Bosse et al., 2006). This finding suggests that the induction of GATA4 expression in tHIO (compare to HIOs) may consolidate anterior intestinal identity promoting the differentiation of CCK-expressing cells, which are uniquely specified in tHIOs. Furthermore, we observed an enrichment of potassium and calcium channel genes in the hormone-producing cells of tHIOs, suggesting functional maturation. Additionally, the proportion of EC cells was higher in HIOs compared to tHIOs, suggesting that this population does not depend on maturation signals or other external cues.

We next inferred differentiation trajectories and lineage-specific regulators of EECs in human tHIOs. Our analysis revealed that peptide hormone-expressing PE cells arise from NEUROG3-late progenitors and subsequently bifurcate into two distinct branches: XMD and LINKSD, which are separate from the EC lineage. Notably, we identified novel cell surface markers of the three human EEC clusters including members of the EPH and integrin families for the XMD cluster, some of which are specific to human EECs.

These findings suggest new hypotheses regarding the pathways that regulate XMD cell differentiation and function, and highlight their interactions with the extracellular matrix. Then, we inferred TF activities dynamically over differentiation trajectories. Interestingly, the correlation between TF expression and activity indicates whether a TF functions as an activator or a repressor. By analyzing the behavior of TFs with known functions in the mouse EEC lineage, our study provides insights into the molecular mechanisms underlying their actions in a given EEC-specific lineage in human intestinal organoids. For instance, FateCompass analysis revealed that in EC trajectories, genes with PAX4 motifs in their promoter are, on average, activated. Thus, PAX4 might control human EC differentiation as a transcriptional activator. When performing differential motif activity analysis to identify lineage-specific regulators dynamically, we identified differentially active transcription factors for the EC, XMD and LINKSD branches that are shared or specific for a given lineage. This suggests, as expected, that common and specialized gene regulatory networks operate to control human EEC cell differentiation. Moreover, our insights into EEC cell differentiation trajectories and drivers provide a framework for generating hypotheses about the specific functions of predicted lineage-specific transcription factors. However, the current data are insufficient to determine which transcription factors can specify whether a cell will produce a given hormone. Addressing this question will require additional replicates, along with information on chromatin state and protein expression at the single-cell level. In addition, functional loss-of-function studies in HIOs are still needed to test their roles in EEC formation. Nevertheless, our findings confirm that HIOs represent a suitable model for investigating human EEC development. Of note, like in the embryonic mouse, relatively advanced differentiated EEC subtypes are generated in HIO and tHIOs despite the fact that these cells are not in direct contact with intestinal lumen, i.e. neither nutrients nor microbiota strengthening the rationale for using HIO/tHIOs in studying EEC biology. The stimuli received by the EECs in tHIOs come mainly from the 'stroma' (blood, muscles, nerves) and are therefore not perceived on the apical side as is the case in the intestine. Interestingly, the amniotic fluid has been suggested to be a source of trophic factors for the developing intestine that could impact EEC development (Dasgupta et al., 2016). Finally, although the processes of intestinal morphogenesis differ between humans and mice – occurring entirely during fetal life in humans and concluding just before weaning in mice – our complementary approaches using mouse embryos and HIOs/tHIOs clearly demonstrate that EEC specification occurs very early during development and follows broadly similar pathways in both species, while retaining species-specific characteristics.

## MATERIALS AND METHODS
### Data collection
#### Animals
Neurog3$^{eYFP}$ mice were housed in an SPF animal facility licensed by the French Ministry of Agriculture (Agreement no. C-67-218-37). The line was maintained on a C57BL/6N background (Charles River). Mice were genotyped by PCR as described previously (Mellitzer et al., 2004).

#### Isolation of mouse EECs and single-cell RNA-sequencing
Heterozygous Neurog3$^{eYFP/+}$ mice were collected at embryonic day E15.5 (*n*=18 embryos) and E18.5 (*n*=29 embryos), and 12 days after birth (P12) (*n*=7 pups). Single cells were isolated from the whole small intestine either directly for stages E15.5 and E18.5, or by first separating the epithelial cells from the rest of the tissue at P12 (Piccand et al., 2019). Intestinal samples were digested into single cells for 30 to 40 min at 37°C in TrypLE™ Express

Biology Open

with vigorous up and downs shakings each 5 min. The cells were centrifuged twice at 900 rpm, 5 min, 4°C in Advanced DMEM, HEPES 10 mM, fetal calf serum (FCS) 10%. For the P12 small intestine, an additional filtering step, onto a 70 µm cell strainer, was performed before centrifugation. Cells were resuspended in Advanced DMEM/F12, HEPES 10 mM, FCS 5%, Y-27632 10 µM, filtered through a 50 µm cell strainer and sorted with a FACS Aria Fusion II (BD) in SORT-seq 384-well cell-capture plates (Single Cell Discoveries). Plates were sealed, centrifuged 2 min at 1500 $g$, 4°C, and snap-frozen and stored at −80°C until sequencing.

Sorted eYFP+ cells were sequenced using the SORT-seq technology (Muraro et al., 2016). Library preparation and single-cell RNA sequencing were performed by Single Cell Discoveries (Utrecht, The Netherlands, https://www.scdiscoveries.com). Demultiplexing, alignment on mm10 assembly of mouse genome, and quantification were performed by Single Cell Discoveries. We ran subsequent analyses using the transcript count matrix and the python-based Scanpy API (Wolf et al., 2018) unless stated otherwise.

### Culture and maintenance of wild-type and NEUROG3-HA-P2A-Venus hiPSC lines

The NEUROG3-HA-P2A-Venus hiPSC line (clone 34, heterozygous) is described in (Schreiber et al., 2021). A 3xHA tag followed by a P2A-3xNLS-Venus cassette is fused to the C-terminus of NEUROG3 on one allele. The parental SB AD3.1 and NEUROG3-HA-P2A-Venus (Schreiber et al., 2021) hiPSC lines were maintained undifferentiated in mTeSR™1 medium (STEMCELL™ Technologies) on hESC grade Matrigel (Corning) as previously described (Schreiber et al., 2021).

### Generation of Human Intestinal Organoids (HIOs)

HIOs were generated using STEMdiff™ Intestinal Organoid Kit (STEMCELL™ Technologies), according to manufacturer's protocol with some modifications. This protocol, originally described by Spence J. and collaborators (Spence et al., 2011), supports the differentiation of hiPSCs into small intestinal organoids, through 3 distinct stages: definitive endoderm (DE), mid-/hindgut (MHG) and small intestine (HIO). Briefly, 24-48 h prior to differentiation, wild-type SB AD3.1 and NEUROG3-HA-P2A-Venus hiPSCs were harvested using TrypLE Select (Gibco™, ThermoFisher Scientific) and seeded at $2×10^5$ cells/well on growth factor reduced-phenol red free Matrigel-coated (Corning) 24-well plates in mTeSR™1 medium (STEMCELL™ Technologies). Differentiation to definitive endoderm (DE) (day 0 to day 2) was started when cells had reached 85-90% confluence. From day 3, DE medium was replaced by MHG medium to induce spheroid formation. Spheroids were collected 2-3 days after exposure to MHG medium, and embedded in Matrigel domes (growth factor reduced-Phenol Red free Matrigel; Corning) to allow their 3D growth, in presence of STEMdiff™ Intestinal organoid growth medium (OGM). Organoids were split every 10-14 days and embedded in new Matrigel domes. Medium was changed every 3-4 days. HIOs were maintained in 3D culture for 31-42 days. For the HIOs that were subsequently transplanted (tHIOs) into immunocompromised NSG mice, the OGM medium was replaced by a medium containing only EGF (Advanced DMEM/F-12, HEPES 10 mM, GlutaMAX 2 mM, penicillin/streptomycin 100 U/ml, supplement B27 1X, supplement N2 1X (all from Gibco™), human recombinant EGF 50 ng/ml (PeproTech®) in order to promote the development of mesenchyme-rich HIOs, essential to ensure good engraftment and growth of the transplant (Múnera et al., 2017; Watson et al., 2014).

### In vivo transplantation of HIOs into the renal subcapsular space of NSG immunocompromised mice

All mice were housed in an animal facility licensed by the French Ministry of Agriculture (Agreement no. C67-218-40), and all animal experiments were approved by the Direction des Services Vétérinaires in compliance with the European legislation on care and use of laboratory animals. The project was authorized by the French Ministry of Higher Education, Research and Innovation. Immunocompromised NOD-SCID IL2rg null (NSG) males were obtained from Charles River France (JAX™ NSG® mice). Transplantations of HIOs under the kidney capsule were carried out as described by Watson and collaborators (Watson et al., 2014) with some

modifications. Antibiotic treatment of mice was reduced to administration in drinking water of trimedoxyne (40 mg/kg/day), 1 day before surgery and then 5 days after transplantation. An analgesic treatment based on buprenorphine (0.1 mg/kg) was administered subcutaneously just before surgery, and the NSAID Meloxicam (1 mg/kg) was given after surgery and in case of pain. The HIOs were maintained in culture for 40-42 days. The Matrigel drop was renewed 24-48 h before transplantation. At the time of transplantation, the HIOs were taken directly from the Matrigel dome (only a thin envelope of Matrigel was left around the HIOs) and one HIO/mouse was grafted under the left kidney capsule as previously described (Watson et al., 2014). The mice were humanely euthanized 15 weeks post-transplantation. Transplants (tHIOs) were then harvested and immediately processed to obtain single cells.

### Dissociation of HIOs and tHIOs into single cells and isolation of EECs for single-cell RNA sequencing

Around 40 HIOs (from a single differentiation) were dissociated to single cells as previously described with some modifications (Grün et al., 2015). Briefly, after medium removal, rinsing of each well with PBS1X, and Matrigel disruption, HIOs were dissociated in TrypLE™ Express (1X) (Gibco™) at 37°C for 20 min, assisted by a mechanical dissociation (up and down with a p1000 pipette). The dissociation was stopped in a basal medium containing Advanced DMEM/F-12, HEPES 10 mM, fetal calf serum 10%, and Y27632 10 µM (STEMCELL™ Technologies). After three washes, the cells were resuspended in the FACS medium (basal medium containing only 5% of fetal calf serum), strained through a 50 µm cell strainer, and stained with DAPI (to exclude dead cells) before being sorted by flow cytometry (FACSAria Fusion, BD) based on Venus fluorescence levels. Individual cells were sorted directly in pre-filled 384-well cell capture plates (one cell/well) containing the reagents to perform single-cell RNA-sequencing (scRNA-Seq) according to the SORT-Seq technology (Muraro et al., 2016) (Single Cell Discoveries).

tHIOs were dissociated to single cells as described by Yu et al. (2021) with modifications. Briefly, a total of eight individual tHIOs were harvested 15 weeks post-transplantation. Samples were first minced into small pieces using a scalpel in ice-cold 1X HBSS, then enzymatically digested in the neural tissue dissociation kit (Miltenyi Biotec, 130-092-628) at 10°C, following the manufacturer's instructions. After complete dissociation, the cells were filtered through a 70 µm cell strainer, centrifuged (500 $g$, 5 min, 10°C) and the pellet was resuspended in the basal medium. The cell suspensions (from 1 to 3 tHIOs, depending on their size) were then labeled using an anti-EpCAM antibody to collect only epithelial cells (APC anti-human CD326 (EpCAM) antibody, BioLegend) and DAPI (to exclude dead cells). Cells were filtered through a 50 µm cell strainer before being sorted by FACS (FACSAria Fusion, BD) directly into the 384-well plates for SORT-Seq analysis (Single Cell Discoveries), as described above. Both DAPI−; EpCAM+; Venus+ or Venus− cells were harvested. Reads were mapped to the human GRCh38 genome assembly.

### Whole-mount immunofluorescent labeling and imaging of HIOs

To characterize the HIOs generated in vitro, whole-mount co-immunostaining with markers of EEC progenitors or differentiated EECs were carried out on whole HIOs. The protocol used was adapted from that described by Dekkers et al. (2019). Briefly, HIOs were recovered from Matrigel domes after 37 days of culture using Cell Recovery Solution (Corning), then fixed in 4% paraformaldehyde (PFA) at 4°C for 45 min. After washing in IF+ buffer (0.2% Triton X-100, 0.05% Tween20, 0.2% Bovine Serum Albumin (BSA) in PBS1X), HIOs were permeabilized for 30 min in 0.5% Triton X-100 in PBS1X at room temperature (RT), rinsed in IF+ buffer for 5 min and incubated in the blocking solution (IF+ buffer, 2% BSA, 5% Donkey Serum) for 30 min at RT, then in the blocking solution supplemented with the primary antibodies (Table S2) overnight (ON) at 4°C with mild rocking. The next day, after several washes in the IF+ buffer, HIOs were incubated in the blocking solution supplemented with the appropriate secondary antibodies, ON at 4°C with mild rocking. The following day, after several washes in IF+ buffer, HIOs were cleared in a fructose-glycerol clearing solution (60% glycerol, 2.5 M fructose) for 30 min at RT, and mounted in the same solution between microscope slide and coverslip

Biology Open

(Dekkers et al., 2019). Slides were imaged using the Spinning Disk, Leica CSU W1 (with 20× dry objective and 25× water immersion objective).

## Immunohistochemistry on tissue sections

Mouse tissues were fixed in 4% paraformaldehyde, PBS at 4°C overnight, cryopreserved in 20% sucrose, PBS at 4°C overnight, and embedded in Sandon Cryomatrix (ThermoFisher Scientific) following classical procedures. Immunostaining on cryo-sections (10 µm thick) was performed using standard protocols. If required, antigen retrieval was performed in 10 mM Sodium Citrate pH6 in a pressure cooker. Primary antibodies are listed in Table S2. Secondary antibodies conjugated to Alexa Fluor® 594, DyLight® 488, DyLight® 549 and DyLight® 649 (Jackson ImmunoResearch) were used at 1:500. For Neurog3, signal amplification was performed using a biotin anti-rabbit (Jackson ImmunoResearch) coupled antibody at 1:500 and streptavidin-Cy3 conjugate at 1:500 (Molecular Probes). Nuclei were stained with DAPI. Imaging was done on Leica DMIRE2 microscope, Leica TCS SP5 Inverted confocal microscope, Spinning disk LEICA CSU W1, or Slide Scanner Hamamatsu, Nanozoomer 2.0 HT. Image analysis was performed with the Fiji software.

## Computational analysis of single-cell RNA-seq data

### Quality control and normalization

To remove low-quality cells, we filtered cells with a high fraction of counts from mitochondrial genes (>20% or 50% in mouse or human data sets respectively). Cells with a high fraction (50% or more) of ERCC spike-in reads were removed from the analysis. Also, cells expressing less than 500 genes in the case of E18.5 and P12 mouse data sets and 3000 genes in the E15.5 data set and 1000 genes in the human data sets were excluded. To account for differences in sequencing depth of cell size, we normalize each cell by the total counts over all genes so that every cell has the same total count after normalization. Finally, 1138 cells and 14,495 genes were kept from the mouse data sets; this output matrix was the input to all further analysis. Quality control and normalization were performed for HIOs and tHIOs samples separately; 633 cells and 16,513 genes were kept for HIOs and 556 cells and 17,511 genes were kept for tHIOs, these output matrices were the input to all further analysis.

### Low dimensional embedding, visualization, and clustering

Principal component analysis was performed using the highly variable genes. A single-cell neighborhood graph using 15 nearest neighbors was computed on the first 15 principal components that sufficiently explain the variation in the data. Batch balanced k nearest neighbors (BBKNN) (Polański et al., 2019) was performed to correct the batch effect among different mouse embryonic and post-natal stages. Uniform manifold approximation and projection (UMAP) (McInnes et al., 2018 preprint) was run for visualization. For clustering and cell-type identification, Louvain-based clustering (Blondel et al., 2008 preprint) at several resolutions in different parts of the manifold was used as implemented in Louvain-igraph in the Scanpy package. Cell types were annotated based on the expression of known marker genes.

### Marker gene identification

Characteristic gene signatures were identified by testing for differential expression of a subgroup against all other cells or between two subgroups as outlined in the text using the Wilcoxon rank-sum method implemented in the *tl.rank_genes_groups* function of Scanpy. One gene was considered as the marker of a group if the FDR was inferior or equal to 0.05, the log-fold change was higher or equal to 1, and it was expressed at least in 15-20% of the cells inside the group, and maximum in 60-65% of the cells outside the group.

### Cell cycle classification

To classify cells into cell cycle phases, we used a cell scoring function described by (Satija et al., 2015) and implemented in the tl.score_genes function in Scanpy with default parameters. The score is the average expression of the gene set subtracted from the average expression of a randomly sampled background set with expression values within the same range. Cell cycle genes to distinguish between S, G2/M, and G1 phase cells were taken from (Tirosh et al., 2016).

## Reconstruction of lineage relationships and pseudotime

To infer lineage relationships between clusters and predict potential differentiation routes, partition-based graph abstraction (PAGA) was performed (Wolf et al., 2019) using the tl.paga function of Scanpy. Edge weights represent the confidence of a connection calculated based on a measure for connectivity. Paths in the PAGA graph signify cluster relationships indicating potential differentiation paths. To infer a pseudotemporal ordering of the cells along the predicted routes in the PAGA graph, diffusion pseudotime (dpt) (Haghverdi et al., 2016) was used as implemented in Scanpy (tl.dpt), setting a root cell within the starting population. The resulting diffusion components were also used for visualization.

## Differentiation trajectories

To outline the differentiation trajectories from the progenitor cells towards the mature-hormone producing cells in mouse EECs, we modeled the system as a discrete Markov process on a network. Assuming that the progenitors behave as sources and the final states as sinks in a dynamical system, the transition probability from one state to another is a function of the potential energy. Then, we performed stochastic simulations following a Monte Carlo sampling algorithm. After 1500 iterations, the simulated differentiation paths were trapped in the respective final states.

## FateCompass specific computations

For downstream analysis of the FateCompass pipeline, we embedded the tHIOs data of gene expression on ten dimensions in the UMAP space. Next, we computed a neighborhood graph in the reduced gene expression space with k=10. This setting was the graph structure for the Markov chain operations of the FateCompass pipeline. The edges of the Markov chain were directed using the potential energy landscape described in Jiménez et al. (2023). FateCompass stochastic trajectories using a Monte Carlo Sampling algorithm. Last, the thresholds for the differential motif activity analysis were: minimum z-score of 1.4, minimum standard deviation over trajectories of 0.004, and minimum Pearson correlation of 0.7.

## Acknowledgements

We acknowledge the animal facilities at IGBMC and ICS for animal care, and William Magnant for his expertise in performing the transplants. We thank the IGBMC Flow cytometry facility for cell sorting, Erwan Grandgirard (Photonic microscopy platform) for help with imaging, Marina Peralta Lopez for editing the video microscopy images and Maulik Narita for help with data submission to GEO.

## Competing interests

The authors declare no competing or financial interests.

## Author contributions

Conceptualization: G.G., S.J., F.B., N.M., A.D.A.; Data curation: G.G., S.J., F.B., N.M., A.D.A.; Formal analysis: S.J., F.B., N.M., A.D.A.; Funding acquisition: G.G., M.M., N.M.; Investigation: A.M., V.S., A.D.A.; Methodology: A.M., V.S., C.G., S.G., M.M.; Project administration: G.G.; Resources: V.S.; Supervision: G.G., V.S., M.M., N.M., A.D.A.; Validation: M.M.; Writing – original draft: S.J., F.B.; Writing – review & editing: G.G., R.K., V.S., A.D.A.

## Funding

This work was funded by the Novo Nordisk Foundation (Challenge Programme Grant NNF14SA0005 to G.G.) and Agence Nationale pour la Recherche (ANR-21-CE14-0003, to G.G., M.M.M. and N.M.). This work of the Interdisciplinary Thematic Institute IMCBio+, as part of the ITI 2021-2028 program of the University of Strasbourg, CNRS and Inserm, was supported by IdEx Unistra (ANR-10-IDEX-0002), and by SFRI-STRAT'US project (ANR-20-SFRI-0012) and EUR IMCBio (ANR-17-EURE-0023) under the framework of the France 2030 Program. Open Access funding provided by the Agence Nationale pour la Recherche (ANR). Deposited in PMC for immediate release.

## Data and resource availability

The raw scRNA-seq data generated in this study are available in the GEO database [GSE 306837]. All other relevant data and details of resources can be found within the article and its supplementary information.

## Peer review history

The peer review history is available online at https://journals.biologists.com/bio/lookup/doi/10.1242/bio.062083.reviewer-comments.pdf

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
