## [Peer Review File · Biology Open]

Unraveling Enteroendocrine Cell lineage dynamics and associated gene regulatory networks during intestinal development

Sara Jimenez, Florence Blot, Aline Meunier, Rishabh Kapoor, Valérie Schreiber, Colette Giethlen, Sabitri Ghimire, Maxime Mahe, Nacho Molina, Adèle De Arcangelis and Gerard Gradwohl

DOI: 10.1242/bio.062083

Editor: Tristan Rodriguez

Review timeline

Original submission: 23 May 2025
First revision: 1 September 2025
Accepted: 4 September 2025

Original submission

First decision letter

MS Title: Unraveling Enteroendocrine Cell lineage dynamics and associated gene regulatory networks during intestinal development

Authors: Gerard Gradwohl; Sara Jimenez; Florence Blot; Aline Meunier; Valérie Schreiber; Colette Giethlen; Sabitri Ghimire; Maxime Mahe; Nacho Molina; Adèle De Arcangelis

Article Type: Research Article

I have now received all the referees' reports on the above manuscript, and have reached a decision. I am sorry to say that the outcome is not a positive one. The referees' comments are appended below, or you can access them online: please go to.

As you will see, the referees raise some significant concerns about your paper, and are not strongly in favour of publication. Having reviewed the critiques and manuscript myself, I understand their concerns and must therefore reject your paper. The main concerns were that the findings were descriptive and largely confirmatory of what has previously been published.

I do realise this is disappointing news, but we receive many more papers than we can publish, and we can only accept manuscripts that receive strong support from referees.

I do hope you find the comments of the referees helpful, and that this decision will not dissuade you from considering our journal for publication of your future work. Many thanks for sending your manuscript to us.

Comments from the Reviewers:

Reviewer 1: SUMMARY OF THE ADVANCE MADE IN THIS PAPER AND ITS POTENTIAL SIGNIFICANCE TO THE FIELD

Jimenez and colleagues use single-cell sequencing approaches to define the developmental differentiation of enteroendocrine cells (EECs), which has not yet been done in mouse or human. They demonstrate that all adult EEC lineages are already present embryonically (in mouse) or in iPSC-derived human intestinal organoids (HIOs) that have been transplanted to mature in a xenograft model (tHIOs). As neither mouse embryos nor HIOs/tHIOs are exposed to (substantial) luminal nutrients or a microbiome, this indicates that EEC differentiation is not dependent on external cues. This is a key finding as it provides evidence that the functional response of EECs to these external cues is independent of their differentiation potential, which is a mechanism that has been widely speculated in the field. The majority of the manuscript is focused on profiling the gene regulatory networks (GRNs) in EECs isolated from tHIOs using a variety of computational models, largely confirming that the transcription factors governing EEC differentiation in tHIOs are similar to established paradigms in mouse. This is excellent work and will be an important resource to the field, however it seems that some of the analysis is rather superficial and these datasets can be further interrogated to shed light on important mechanisms governing EEC differentiation. I have a few concerns and suggestions that may improve the manuscript.

SUGGESTIONS TO AUTHORS

Major:

1. Important experimental details are not reported, including number of samples analyzed or whether samples were pooled from multiple biological replicates (mouse, HIO, and tHIO). It was also unclear whether the P12 mouse intestine included all regions of the small intestine (as done for E15.5 and E18.5 embryos).
2. At this resolution, EECs are broadly divided into enterochromaffin (EC) and peptidergic (PE) clusters. While some hormones are quite restricted to certain clusters (eg GIP in mouse KIS, GHRL/MLN in HIO XMD-LK), many hormones (GCG, SST) are expressed across multiple clusters, albeit at variable levels. There is a large step missing from the analysis presented here that connects the GRNs identified to final hormonal expression. Can the present datasets be used to fill this gap? Are transcription factors sufficient to determine whether a cell will produce a given hormone? If the present dataset is insufficient for answering this question, could the authors discuss the next steps to address this gap?
3. Figure 1: There looks to be a shift in the progenitor populations between embryonic stages and P12 (potentially coinciding with crypt development?). With this in mind, it would be useful to see the Pseudotime analysis at all 3 timepoints rather than only the merged data.
4. Figure 1: Are any of these progenitor populations proliferative at any timepoint? It would be nice to see the cell cycle plot consistent with the ones shown for HIOs and tHIOs.
5. HIO Methods: The description of the medias used in the 31-42 day 3D growth phase for HIOs is a little confusing. It seems like the HIOs analyzed in vitro received OGM whereas the HIOs set aside for transplant received EGF. If this is accurate, care should be taken when comparing the HIO and tHIO datasets, as in vivo maturation is not the only variable that may affect gene expression. For example, is the high ASCL1 expression in HIOs in vitro due to the media composition, the reduced mesenchymal population, or a true representation of residual "pluripotency" as suggested in the text?
6. A key missing analysis and discussion surrounds the "molecular mechanism" (line 301) that enables the maturation of EECs in HIOs after transplant. Do these datasets provide clues into the genes that separate XMD-LK cells in vitro to XMD and LINKSD cells after transplant?
7. While interesting, the data shown in Figure 6 could be moved to the supplement. Alternatively, did your analysis uncover novel cell surface markers of EC or LINKSD cells? This would make a more compelling main figure. I would also suggest keeping the nomenclature consistent throughout the manuscript and refer to these cells as XMD cells, not "Ghrelin cells" (line 994).
8. Figure 7 (lines 336-351): The description of the populations emerging "early" versus "late" is confusing. Is the interpretation of this analysis that LINKSD cells remain a progenitor for longer, whereas XMD and EC cells differentiate with less time spent as a progenitor?
9. Figure 8 (lines 353-387): The description of the activity analysis is also a little confusing. What were the target genes used for this analysis? Was this analysis only done for the 3 transcription factors demonstrated in Figure 8 (ARX, PAX4, LMX1A)? Is the interpretation that ARX made in XMD

and LINKSD cells regulates expression of target genes in progenitor populations? And that LMX1A made in late ECs regulates gene expression cell autonomously in late EC cells as well as regulating expression in LINKSD cells? More in-depth analysis and discussion could uncover some important molecular mechanisms by which EECs regulate each other, thus maintaining consistent pools of hormone expression, as well as factors which drive the differentiation of distinct EEC cell populations. Also, the legend for Figure 8 should remain consistent with the nomenclature used throughout the paper (EC, XMD, and LINKSD rather than specific hormones).

10. Figure 9: The organization of this figure is messy and confusing. I would suggest lining up all these plots into the three columns of cell populations (akin to a heatmap where genes not participating in a given trajectory should be all blue). I think the genes should still be legible to fit on one page.

11. Figures 8/9: I would suggest performing the same activity analysis as done in Figure 8 for the transcription factors highlighted in Figure 9. This additional analysis could be more clearly depicted if figures 8 and 9 were switched, and the "functional" activity analysis for these key genes is the final figure of the paper. This could help build a map of GRNs regulating the differentiation of these populations.

12. I strongly suggest including a model synthesizing the key data found with these analyses - such as a map of GRNs regulating the differentiation of these populations.

Minor

1. Figure 2: I suggest either combining this experimental design with Figure 3 or moved to the supplement, as HIOs and tHIOs have been described and cited thoroughly in the text.

2. Figure 4D should be also represented as a dot plot to be consistent with the mouse and HIO datasets.

3. As described by others in the mouse (Schonhoff et al 2004, Enriquez et al 2022), lineage tracing using Neurog3 gives rise to sporadic "ribbons" consisting of all intestinal epithelial cells. Did the authors observe any labeled ribbons in the tHIOs?

4. Figure 5 (line 296): Sinagoga et al 2018 demonstrated that HIOs prior to transplantation were already functional by hormone secretion in response to luminal glucose.

5. Figure 7/Figure 8: The legend color (gray) does not match the graph color (dark blue) for the LINKSD cluster.

6. Discussion (line 486): How does this "new" differentiation branch differ from the polyhormonal cells described in Egerod et al 2012?

7. Discussion (line 534): Here is an opportunity to connect the mouse data in Figure 1 with the HIO data in the rest of the paper. Like the embryonic mouse, HIOs and tHIOs do not have luminal nutrients or a microbiome and they still make the full complement of EECs. Highlighting this point will strengthen the rationale for using HIOs/tHIOs in studying EEC biology.

8. Typos: Table B11 (unknown), Figure S5D (unknown), line 1016 (TPH1)

Reviewer 2: SUMMARY OF THE ADVANCE MADE IN THIS PAPER AND ITS POTENTIAL SIGNIFICANCE TO THE FIELD

The authors present an EEC analysis during the early stages of intestinal development between mouse and human model systems. The paper exclusively relies upon single-cell analysis and predictive modeling in mouse or human intestine (in-vitro HIOs & transplanted HIOs). The paper concludes by presenting a predicted gene regulatory network of various EEC lineages during development as they claim, "before crypt-villus organization, diet or microbiota." Overall, the paper seems promising and focuses on an understudied area of EEC differentiation. Current published works focus on the adult timeframe or overexpression of Neurog3, but this paper seeks to advance EEC GRNs during early development. The paper is written in an easy to follow manner and presentation of data is digestible. However, although some novel ideas are presented, a more thorough analysis of data is needed as well as more explanations of biological significance. Although, the tools utilized for computational analysis are repetitive during initial Fig 1-4, the more potentially significant findings are shown in Fig 5-9. These figures may be reduced and combined to emphasize cohesion of ideas. Furthermore, to aid the utility of their predictive modeling, the paper will benefit from validation experiments of some genes that are highlighted. Revisions to data analysis, attenuation of text related to EEC development relying on environmental cues

© 2025. Published by The Company of Biologists under the terms of the Creative Commons Attribution License (<https://creativecommons.org/licenses/by/4.0/>).

(crypt/villus signaling, diet, microbiome), and extending their discussion to incorporate significance of potentially modulating developmental cues, will further improve this paper for publication. Significant revisions are recommended to text and data analysis.

SUGGESTIONS TO AUTHORS

- Crypt/villus formation occurs during gestation in humans but postnatally in mice. Perhaps, further inquiry into comparisons of their data at mouse P12 vs transplanted- HIOs could aid in describing differences during EEC differentiation/specification.
- For mouse data, due to lack of crypts and/or distinct Paneth cell population, how does lack of paracrine factors affect differentiation of EECs during development?
- Motif analysis is performed but placed in supplement. Moving that data to main figure with accompanying gene ontology analysis may help emphasize role/significance in EEC differentiation bifurcation stages.
- A schematic "roadmap" of novel GRNs discovered during single-cell analysis will enable visualization of impact of data (perhaps coloring new transcription factors or other regulatory genes distinct from previously published genes will help)
- Clarification of source (n-value) of single cell data needed. For mouse, how many mice were used, male or female? For HIOs and tHIOs, were these combined differentiations or from 1 batch?
- For all single cell data, what region of the intestine is analyzed? Seems to be proximal focused. Generation of a gene score system utilizing genes that depict regionality (Hox genes or genes regulated by CDX2) can help identify regionality and or differences in EEC differentiation based on intestinal region.
- Related to novel genes identified (EPHA4, ITGB8, ITGA7, etc.)- perform RNA-scope or other validation study to complement single-cell analysis. For example, it may be helpful to knockdown or find knockout models of EPHA4 (or other promising gene candidate) to determine if any effects on EEC specification.
- Comparison of data to Neurog3^{-/-} or other related EEC regulatory transcription factors may further highlight novel genes during development and establish GRN.
- Are there enough cells/data from single cell experiments to determine the role of Neurog3 gene dosage in delineating EECs? In adult context, Neurog3 dosage affects secretory cell fate. Does this also occur during development?
- Transplanted HIOs engraft to the kidney capsule and most likely deliver nutrients to the organoid. Could this not only enable maturation but also turn on GRNs relevant for EEC differentiation?
- What is the effect of the amniotic fluid during development on EECs? There are studies in rabbit and mice whereby fetal intestinal development is impaired in changes to amniotic fluid composition. Perhaps, nutrients within the fluid are relevant to intestinal development and could contribute to EEC development/GRN activation.
- What is the biological significance of so many progenitor populations/overlapping hormonal populations? Could other environment cues be required to delineate distinct hormone populations?
- Antibiotic exposure during embryonic development and maternal diet influences fetal intestinal development. Authors should expand in discussion how their data fits with previously published hypotheses.
- Expand in discussion on potential differences of GRNs regulating differentiation within other populations during developmental period (Enterocyte, goblet, paneth, stem cell)
- For all tables in supplement, a heatmap should be used to summarize the top 50 genes

Reviewer 3: Dear Dr. Gradwahl and colleagues,

This is an interesting study, with some important findings and hypotheses. The core of the study is single-cell RNAseq data from mouse embryos and neonates (P12), HIOs and transplanted HIOs. The study shows that the embryonic and adult development are similar, and that HIOs follow the differentiation program from the literature. Moreover, transplanted HIOs have a more "mature" program. These are all important confirmatory results that establish the HIOs and mice as valid model systems. You also make an important observation, that while nutrition and microbiota were

© 2025. Published by The Company of Biologists under the terms of the Creative Commons Attribution License (<https://creativecommons.org/licenses/by/4.0/>).

implied as important for EEC differentiation, the differentiation program is very similar in HIOs and in embryonic stages, suggesting that the role of these external factors is secondary. The simulations and analyses of differentiation programs also highlight some new factors that may be important for specific lineages of enteroendocrine differentiation and maturation. I believe these are all good points. The manuscript is written well and easy to follow and understand its main takes. I am not sure how to interpret the neg correlations in some lineages in figure 8 and this was not referred to in the text (e.g arx in 2 lineages).

I have few concerns:

First, the data is very descriptive. There are many maps describing pseudotimes and UMAPs, but no perturbations, either from data or from simulations (e.g. how would a loss of a TF affect differentiation or ratios of cells? maybe link it to a phenotype in the literature).

Second, the data is nearly completely at the RNA level. All new hypotheses on TFs are based on analyses, without any validations at the protein level. We know there is actually very low correlation between RNA and protein levels, and scRNAseq is even worse, so I would advise that the authors take some central findings and identify the proteins using proteomics/westerns/immunostaining/... There is also no physiological tests on the tHIOs or HIOs (hormone levels). How can you make a claim on maturation?

Third - I didn't see a formal analysis on the differences between embryonic-neonatal-adult mouse EECs.

Finally, I really liked the insight that EECs develop in the gut without microbiota/nutrition/mechanical stimulation and mature in the kidney capsule without interaction with the GI environment. This suggests maybe that the effects of the GI environment is secondary to the maturation that occurs to all(?) stem-cell derived organoids transplanted to the kidney capsule. Can find what is it in the GI environment that affects differentiation/function and what is simply the adult environment? There may be some databases on germ-free EECs maybe. This is an important insight.

Good luck

Author response to reviewers' comments

Reviewer 1: SUMMARY OF THE ADVANCE MADE IN THIS PAPER AND ITS POTENTIAL SIGNIFICANCE TO THE FIELD

Jimenez and colleagues use single-cell sequencing approaches to define the developmental differentiation of enteroendocrine cells (EECs), which has not yet been done in mouse or human. They demonstrate that all adult EEC lineages are already present embryonically (in mouse) or in iPSC-derived human intestinal organoids (HIOs) that have been transplanted to mature in a xenograft model (tHIOs). As neither mouse embryos nor HIOs/tHIOs are exposed to (substantial) luminal nutrients or a microbiome, this indicates that EEC differentiation is not dependent on external cues. This is a key finding as it provides evidence that the functional response of EECs to these external cues is independent of their differentiation potential, which is a mechanism that has been widely speculated in the field. The majority of the manuscript is focused on profiling the gene regulatory networks (GRNs) in EECs isolated from tHIOs using a variety of computational models, largely confirming that the transcription factors governing EEC differentiation in tHIOs are similar to established paradigms in mouse. This is excellent work and will be an important resource to the field, however it seems that some of the analysis is rather superficial and these datasets can be further interrogated to shed light on important mechanisms governing EEC differentiation. I have a few concerns and suggestions that may improve the manuscript.

SUGGESTIONS TO AUTHORS

Major:

© 2025. Published by The Company of Biologists under the terms of the Creative Commons Attribution License (<https://creativecommons.org/licenses/by/4.0/>).

1. Important experimental details are not reported, including number of samples analyzed or whether samples were pooled from multiple biological replicates (mouse, HIO, and tHIO). It was also unclear whether the P12 mouse intestine included all regions of the small intestine (as done for E15.5 and E18.5 embryos).

Answer: The requested information has now been incorporated into the Material and methods section or in the figure legends. The P12 mouse intestine samples include the whole small intestine, but only the epithelial cells were analyzed.

2. At this resolution, EECs are broadly divided into enterochromaffin (EC) and peptidergic (PE) clusters. While some hormones are quite restricted to certain clusters (eg GIP in mouse KIS, GHRL/MLN in HIO XMD-LK), many hormones (GCG, SST) are expressed across multiple clusters, albeit at variable levels. There is a large step missing from the analysis presented here that connects the GRNs identified to final hormonal expression. Can the present datasets be used to fill this gap? Are transcription factors sufficient to determine whether a cell will produce a given hormone? If the present dataset is insufficient for answering this question, could the authors discuss the next steps to address this gap?

Answer: The current data are insufficient to determine which transcription factors are sufficient to specify whether a cell will produce a given hormone. Answering this question would need more replicates, as well as information on chromatin state and protein expression at the single cell level. This point is now discussed in the revised manuscript (line 541-546).

3. Figure 1: There looks to be a shift in the progenitor populations between embryonic stages and P12 (potentially coinciding with crypt development?). With this in mind, it would be useful to see the Pseudotime analysis at all 3 timepoints rather than only the merged data.

Answer: Yes there is a shift in the progenitor population at P12, where progenitors represent 30% of the cells at this stage and are actively cycling (see also response point 4). This information has been added in the Results section (line 115-117). Additionally we have included a new figure (Figure S1) showing the pseudotime analysis at all 3 timepoints, as well as merged data, as requested.

4. Figure 1: Are any of these progenitor populations proliferative at any timepoint? It would be nice to see the cell cycle plot consistent with the ones shown for HIOs and tHIOs.

Answer: We have added a new panel to figure 1 (panel C) showing cycling cells, which we found to be predominantly restricted to progenitor cells of the P12 stage. This information has been added in the Results section (line 115-117).

5. HIO Methods: The description of the medias used in the 31-42 day 3D growth phase for HIOs is a little confusing. It seems like the HIOs analyzed in vitro received OGM whereas the HIOs set aside for transplant received EGF. If this is accurate, care should be taken when comparing the HIO and tHIO datasets, as in vivo maturation is not the only variable that may affect gene expression.

Answer: The observation made by reviewer 1 is accurate. Accordingly we have added a sentence, at the end of the paragraph describing these results, to acknowledge this limitation in interpreting the comparison between the HIO and tHIO datasets (line 306-308).

6. A key missing analysis and discussion surrounds the "molecular mechanism" (line 301) that enables the maturation of EECs in HIOs after transplant. Do these datasets provide clues into the genes that separate XMD-LK cells in vitro to XMD and LINKSD cells after transplant?

Answer: The current data do not permit identification of the molecular mechanisms underlying EEC maturation following transplantation of HIOs, nor do they explain the divergence of XMD-LK cells into XMD and LINKSD subtypes. Progress in this area will require analysis of a larger

number of cells, along with assessment of changes in chromatin accessibility and transcription factor binding. This point is briefly discussed in the discussion.

7. While interesting, the data shown in Figure 6 could be moved to the supplement. Alternatively, did your analysis uncover novel cell surface markers of EC or LINKSD cells? This would make a more compelling main figure. I would also suggest keeping the nomenclature consistent throughout the manuscript and refer to these cells as XMD cells, not "Ghrelin cells" (line 994).

Answer: We have followed Reviewer 1's suggestion and extended the search for cell surface markers to EC and LINKSD cells. As a result, we have included a modified Figure 6 including these data and described these findings in the corresponding paragraph of the Results section, which is now titled: 'Cell surface markers of human EECs'.

8. Figure 7 (lines 336-351): The description of the populations emerging "early" versus "late" is confusing. Is the interpretation of this analysis that LINKSD cells remain a progenitor for longer, whereas XMD and EC cells differentiate with less time spent as a progenitor?

Answer: Our interpretation is that for the LINKSD fate, intermediate states may be rare or transient making them more difficult to capture. This clarification is now stated on lines 401-402. In addition, regarding the emergence of the different fates, we mention later in the manuscript that " Pseudotime analysis suggested that XMD and LINKSD arise at similar moments; and as expected, EC early cells start to differentiate before EC late cells." (lines 255-256).

9. Figure 8 (lines 353-387): The description of the activity analysis is also a little confusing. What were the target genes used for this analysis? Was this analysis only done for the 3 transcription factors demonstrated in Figure 8 (ARX, PAX4, LMX1A)? Is the interpretation that ARX made in XMD and LINKSD cells regulates expression of target genes in progenitor populations? And that LMX1A made in late ECs regulates gene expression cell autonomously in late EC cells as well as regulating expression in LINKSD cells? More in-depth analysis and discussion could uncover some important molecular mechanisms by which EECs regulate each other, thus maintaining consistent pools of hormone expression, as well as factors which drive the differentiation of distinct EEC cell populations. Also, the legend for Figure 8 should remain consistent with the nomenclature used throughout the paper (EC, XMD, and LINKSD rather than specific hormones).

Answer: To validate the use of Fatecompass in studying EEC differentiation in tHIOs, Figure 8 first presents the activity dynamics of ARX, PAX4 and LMX1A, whose roles in EEC differentiation has been characterized at least in the mouse. This analysis was extended to all TF with known binding motifs and the global results are shown in Figure 9 as heatmaps.

In Figure 9, to identify lineage-specific regulators dynamically, we leverage the differential motif activity analysis of FateCompass. We defined a differential motif activity analysis based on the following criteria: (i) motifs with the high positive z- score, i.e., motifs that significantly varied across cells compared with their estimated errors; (ii) motifs with high activity variability across the lineage-specific differentiation trajectory. Finally, (iii) motifs with a high temporal correlation between its activity and mRNA expression within a specific window of time lags as described in Jiménez et al., Cell Reports Methods 2023. These clarifications have been incorporated in the manuscript (line 400-404).

ARX expression is weak in progenitors but increases with time both in the LINKS and XMD clusters. ARX activity is the strongest in the XMD lineage but declines in the LINKSD so it could be that indeed ARX targets genes in progenitors to determine cell fate and others in more differentiated XMD cells to initiate functional programs in these cells. For LMX1A, we interpret the activity observed in LINKSD cells as reflecting genes repressed by LMX1A or genes regulated by other TFs sharing LMX1A binding motif.

The legend of figure 8 has been modified.

10. Figure 9: The organization of this figure is messy and confusing. I would suggest lining up all these plots into the three columns of cell populations (akin to a heatmap where genes not participating in a given trajectory should be all blue). I think the genes should still be legible to fit on one page.

Answer: We agree, Figure 9 has been reorganized in 3 columns (one column per cell cluster) and all the genes are now shown providing readers with a clearer and more comprehensive overview of the data.

11. Figures 8/9: I would suggest performing the same activity analysis as done in Figure 8 for the transcription factors highlighted in Figure 9. This additional analysis could be more clearly depicted if figures 8 and 9 were switched, and the "functional" activity analysis for these key genes is the final figure of the paper. This could help build a map of GRNs regulating the differentiation of these populations.

Answer: We prefer to keep the order of Figures 8 and 9 as it is. Indeed, as explained in point 9, Figure 8 is dedicated to validating the use of FateCompass by characterizing the dynamics of transcription factor (TF) activity with known roles in EEC differentiation, whereas Figure 9 presents the differential motif activity based on specific criteria (see point 9). As suggested, we have now included an additional graph depicting the activity profiles along differentiation trajectories for a selection of TF activities identified in Figure 9. We have also shortened the paragraph of this section "Transcriptional interactions underlying enteroendocrine subtype specification in tHIOs" to reduce manuscript length (removed the description of FOXA3). These results are presented in Supplementary Figure S11.

12. I strongly suggest including a model synthesizing the key data found with these analyses - such as a map of GRNs regulating the differentiation of these populations.

Answer: We agree that a map of the GRNs regulating the differentiation of the EEC population would indeed be very valuable. However, we feel that the data currently available are still too limited to support such a model with confidence. In addition, FateCompass identified 133 differentially active TFs, which makes it challenging to concisely summarize the findings. To address this, the revised Figure 9 now presents the complete list of these TFs along with their lineage specific activity, providing readers with a clearer and more comprehensive overview of the data. For these reasons and to avoid the risk of presenting an oversimplified or potentially misleading schematic, we prefer not to include one at this stage. We hope that future studies, building on our findings, will make it possible to construct such a model with greater accuracy.

Minor

1. Figure 2: I suggest either combining this experimental design with Figure 3 or moved to the supplement, as HIOs and tHIOs have been described and cited thoroughly in the text.

Answer: We prefer to keep Figure 2 as a separate figure (since there is not enough space to incorporate it into Figure 3), as it provides the reader with a clear schematic overview of the experimental design.

2. Figure 4D should be also represented as a dot plot to be consistent with the mouse and HIO datasets.

Answer: The dot plot with the markers for all the sequenced cells is now shown in Figure 4B.

3. As described by others in the mouse (Schonhoff et al 2004, Enriquez et al 2022), lineage tracing using Neurog3 gives rise to sporadic "ribbons" consisting of all intestinal epithelial cells. Did the authors observe any labeled ribbons in the tHIOs?

© 2025. Published by The Company of Biologists under the terms of the Creative Commons Attribution License (<https://creativecommons.org/licenses/by/4.0/>).

We never observed ribbons of venus+ cells in the tHIOs

4. Figure 5 (line 296): Sinagoga et al 2018 demonstrated that HIOs prior to transplantation were already functional by hormone secretion in response to luminal glucose.

Answer: This information has been added to the manuscript (line 299-300).

5. Figure 7/Figure 8: The legend color (gray) does not match the graph color (dark blue) for the LINKSD cluster.

Answer: fixed

6. Discussion (line 486): How does this "new" differentiation branch differ from the polyhormonal cells described in Egerod et al 2012?

Answer: This is difficult to assess, as in Egerod et al. the analyzed EEC population consisted of sorted CCK-GFP-positive cells, and RNA-seq data for this population are not available.

7. Discussion (line 534): Here is an opportunity to connect the mouse data in Figure 1 with the HIO data in the rest of the paper. Like the embryonic mouse, HIOs and tHIOs do not have luminal nutrients or a microbiome and they still make the full complement of EECs. Highlighting this point will strengthen the rationale for using HIOs/tHIOs in studying EEC biology.

Answer: We thank reviewer 1 for this suggestion and have added this comment in the discussion (line 546-547).

8. Typos: Table B11 (unknown), Figure S5D (unknown), line 1016 (TPH1)

Answer: fixed

Reviewer 2: SUMMARY OF THE ADVANCE MADE IN THIS PAPER AND ITS POTENTIAL SIGNIFICANCE TO THE FIELD

The authors present an EEC analysis during the early stages of intestinal development between mouse and human model systems. The paper exclusively relies upon single-cell analysis and predictive modeling in mouse or human intestine (in-vitro HIOs & transplanted HIOs). The paper concludes by presenting a predicted gene regulatory network of various EEC lineages during development as they claim, "before crypt-villus organization, diet or microbiota." Overall, the paper seems promising and focuses on an understudied area of EEC differentiation. Current published works focus on the adult timeframe or overexpression of Neurog3, but this paper seeks to advance EEC GRNs during early development. The paper is written in an easy to follow manner and presentation of data is digestible. However, although some novel ideas are presented, a more thorough analysis of data is needed as well as more explanations of biological significance. Although, the tools utilized for computational analysis are repetitive during initial Fig 1-4, the more potentially significant findings are shown in Fig 5-9. These figures may be reduced and combined to emphasize cohesion of ideas. Furthermore, to aid the utility of their predictive modeling, the paper will benefit from validation experiments of some genes that are highlighted. Revisions to data analysis, attenuation of text related to EEC development relying on environmental cues (crypt/villus signaling, diet, microbiome), and extending their discussion to incorporate significance of potentially modulating developmental cues, will further improve this paper for publication. Significant revisions are recommended to text and data analysis.

SUGGESTIONS TO AUTHORS

- Crypt/villus formation occurs during gestation in humans but postnatally in mice. Perhaps,

© 2025. Published by The Company of Biologists under the terms of the Creative Commons Attribution License (<https://creativecommons.org/licenses/by/4.0/>).

further inquiry into comparisons of their data at mouse P12 vs transplanted- HIOs could aid in describing differences during EEC differentiation/specification.

Answer: We thank Reviewer 2 for this suggestion. Unfortunately we believe such a study would not be conclusive given the relatively low number of cells sorted and sequenced at P12 compared to HIOs.

- For mouse data, due to lack of crypts and/or distinct Paneth cell population, how does lack of paracrine factors affect differentiation of EECs during development?

Answer: In mice, EEC development in the embryo resembles the differentiation of adult EECs (this study) suggesting that crypts, Paneth cells are not essential.

- Motif analysis is performed but placed in supplement. Moving that data to main figure with accompanying gene ontology analysis may help emphasize role/significance in EEC differentiation bifurcation stages.

Answer: Now, in the modified Figure 9 the complete motifs analysis is shown and thus the table in the supplement has been removed.

- A schematic "roadmap" of novel GRNs discovered during single-cell analysis will enable visualization of impact of data (perhaps coloring new transcription factors or other regulatory genes distinct from previously published genes will help)

Answer: We fully agree that a map of the GRNs regulating the differentiation of the EEC population would be of great value. However, we feel that the data currently available are still too limited to support such a model with confidence. In addition, FateCompass identified 133 differentially active TFs, which makes it challenging to concisely summarize the findings. To address this, the revised Figure 9 now presents the complete list of these TFs along with their lineage specific activity, providing readers with a clearer and more comprehensive overview of the data. For these reasons and to avoid the risk of presenting an oversimplified or potentially misleading schematic, we prefer not to include one at this stage. We hope that future studies, building on our findings, will make it possible to construct such a model with greater accuracy.

- Clarification of source (n-value) of single cell data needed. For mouse, how many mice were used, male or female? For HIOs and tHIOs, were these combined differentiations or from 1 batch?

Answer: this information has been added in the Material and Methods section.

- For all single cell data, what region of the intestine is analyzed? Seems to be proximal focused. Generation of a gene score system utilizing genes that depict regionality (Hox genes or genes regulated by CDX2) can help identify regionality and or differences in EEC differentiation based on intestinal region.

Answer: For the mouse, we analyzed cells from the small intestine, now indicated in the Material and Methods. The strong expression of GATA4 suggests that as expected tHIOs have a proximal intestinal identity.

- Related to novel genes identified (EPHA4, ITGB8, ITGA7, etc.)- perform RNA-scope or other validation study to complement single-cell analysis. For example, it may be helpful to knockdown or find knockout models of EPHA4 (or other promising gene candidate) to determine if any effects on EEC specification.

Answer: EECs are a rare population, and the tools required to address this question (e.g., probes or antibodies) need thorough validation. Therefore, confirming our findings by RNAscope or immunofluorescence for novel genes may be technically very challenging. Validation of the

identified genes through loss-of-function studies is beyond the scope of this manuscript. Nevertheless, as our results suggested that VDR may play a role in XMD cell differentiation (Figure 9) we tested this hypothesis. We had access to VDR knockout mice and analyzed Ghrelin expression, but found that the formation of Ghrelin cells was not altered in the VDR-deficient intestine. For this reason, we decided not to include these data. This result however does not preclude VDR to play a role in Ghrelin cells differentiation in the intestine.

- Comparison of data to Neurog3^{-/-} or other related EEC regulatory transcription factors may further highlight novel genes during development and establish GRN.

Answer: We thank Reviewer 2 for this suggestion but we do not have bulk or scRNAseq data of Neurog3^{-/-} (mouse or HIOs) in hand, or of other EEC TF.

- Are there enough cells/data from single cell experiments to determine the role of Neurog3 gene dosage in delineating EECs? In adult context, Neurog3 dosage affects secretory cell fate. Does this also occur during development?

Answer: Unfortunately addressing this question would require quantification of Neurog3 protein as well at the single cell level and trace these cells to determine whether Neurog3 levels affect their destiny which, to our knowledge is technically currently impossible.

- Transplanted HIOs engraft to the kidney capsule and most likely deliver nutrients to the organoid. Could this not only enable maturation but also turn on GRNs relevant for EEC differentiation?

Answer: We agree, that is why we mention in the discussion that "The stimuli received by the EECs in tHIOs come mainly from the "stroma" (blood, muscles, nerves)" which includes the site of transplantation and it is possible that these nutrients promote EEC differentiation.

- What is the effect of the amniotic fluid during development on EECs? There are studies in rabbit and mice whereby fetal intestinal development is impaired in changes to amniotic fluid composition. Perhaps, nutrients within the fluid are relevant to intestinal development and could contribute to EEC development/GRN activation.

Answer: This hypothesis has now been mentioned in the discussion (line 551)

- What is the biological significance of so many progenitor populations/overlapping hormonal populations? Could other environment cues be required to delineate distinct hormone populations?

Answer: The co-expression of multiple hormones within a single enteroendocrine cell may provide functional plasticity, allowing the cell to secrete different combinations of hormones in response to diverse stimuli. Beumer et al, 2018 have shown that a BMP gradient along the crypt-villi axis regulates hormone switching in migrating EECs (mentioned in the introduction)

- Antibiotic exposure during embryonic development and maternal diet influences fetal intestinal development. Authors should expand in discussion how their data fits with previously published hypotheses.

Answer: It is not to entirely clear which results Reviewer 2 is referring to. To our knowledge there is no published study on the effect of antibiotic treatment or maternal diet on the differentiation of EEC.

- Expand in discussion on potential differences of GRNs regulating differentiation within other populations during developmental period (Enterocyte, goblet, paneth, stem cell)

Answer: Our manuscript focuses on the gene regulatory networks controlling EEC differentiation; we therefore believe it should remain centered on this core topic

- For all tables in supplement, a heatmap should be used to summarize the top 50 genes

Answer: Thank you for this suggestion, as requested we changed the tables showing the top 50 differentially expressed genes in each cluster for HIO (Figures S4 and S5) and tHIO by heatmaps (Figures S8 and S9).

Reviewer 3: Dear Dr. Gradwohl and colleagues,

This is an interesting study, with some important findings and hypotheses. The core of the study is single-cell RNAseq data from mouse embryos and neonates (P12), HIOs and transplanted HIOs. The study shows that the embryonic and adult development are similar, and that HIOs follow the differentiation program from the literature. Moreover, transplanted HIOs have a more "mature" program. These are all important confirmatory results that establish the HIOs and mice as valid model systems. You also make an important observation, that while nutrition and microbiota were implied as important for EEC differentiation, the differentiation program is very similar in HIOs and in embryonic stages, suggesting that the role of these external factors is secondary.

The simulations and analyses of differentiation programs also highlight some new factors that may be important for specific lineages of enteroendocrine differentiation and maturation. I believe these are all good points. The manuscript is written well and easy to follow and understand its main takes. I am not sure how to interpret the neg correlations in some lineages in figure 8 and this was not referred to in the text (e.g arx in 2 lineages).

I have few concerns:

First, the data is very descriptive. There are many maps describing pseudotimes and UMAPs, but no perturbations, either from data or from simulations (e.g. how would a loss of a TF affect differentiation or ratios of cells? maybe link it to a phenotype in the literature).

Answer: The primary goal of this study was to generate novel hypotheses regarding the function of predicted lineage transcription factors in human EEC differentiation. While perturbation experiments, such as transcription factor loss-of-function studies, would indeed be valuable, they are beyond the scope of the present manuscript and could be addressed in future work. Nevertheless, as suggested by Reviewer 3 we have added additional references to published data supporting the predicted GRN regulating human EEC development (e.g PROX1, SNAI1) lines 443-449.

Second, the data is nearly completely at the RNA level. All new hypotheses on TFs are based on analyses, without any validations at the protein level. We know there is actually very low correlation between RNA and protein levels, and scRNAseq is even worse, so I would advise that the authors take some central findings and identify the proteins using proteomics/westerns/immunostaining/... There is also no physiological tests on the tHIOs or HIOs (hormone levels). How can you make a claim on maturation?

Answer: EECs are a rare population, and the tools required to address this question (e.g., probes or antibodies) need thorough validation. Therefore, confirming our findings by RNAscope or immunofluorescence for novel genes may be technically very challenging. Validation of the identified genes through loss-of-function studies is beyond the scope of this manuscript. Nevertheless, as our results suggested that VDR may play a role in XMD cell differentiation (Figure 9) we tested this hypothesis.

We had access to VDR knockout mice and analyzed Ghrelin expression, but found that the formation of Ghrelin cells was not altered in the VDR-deficient intestine. For this reason, we decided not to include these data. This result however does not preclude VDR to play a role in

Ghrelin cells differentiation in the intestine. When we refer to maturation, we either specifically mean 'molecular' signs of EEC maturation, or intestinal tissue maturation (crypt/villi formation). We have carefully reviewed the text to either remove the term or rephrase it to avoid overstatement.

Third - I didn't see a formal analysis on the differences between embryonic-neonatal-adult mouse EECs.

Finally, I really liked the insight that EECs develop in the gut without microbiota/nutrition/mechanical stimulation and mature in the kidney capsule without interaction with the GI environment. This suggests maybe that the effects of the GI environment is secondary to the maturation that occurs to all(?) stem-cell derived organoids transplanted to the kidney capsule. Can find what is it in the GI environment that affects differentiation/function and what is simply the adult environment? There may be some databases on germ-free EECs maybe. This is an important insight.

Answer: We did not perform a formal comparison between embryonic/neonatal and adult mouse EECs, as their differentiation trajectories and cellular diversity appear largely similar. Moreover, the primary focus of this study is on human EECs.

Second decision letter

MS ID#: bio.062083R1

MS Title: Unraveling Enteroendocrine Cell lineage dynamics and associated gene regulatory networks during intestinal development

Authors: Gerard Gradwohl; Sara Jimenez; Florence Blot; Aline Meunier; Valérie Schreiber; Colette Giethlen; Sabitri Ghimire; Maxime Mahe; Nacho Molina; Adèle De Arcangelis

Article Type: Research Article

I am happy to tell you that your manuscript has been accepted for publication in Biology Open, pending our standard publication integrity checks. It was accepted on 4th September 2025.